# Nitrogen fixation under declining Arctic sea ice
Lisa W. von Friesen [1,11], Hanna Farnelid[2], Wilken-Jon von Appen[3], Mar Benavides [4,5,6], Olivier Grosso[4], Christien P. Laber[2,12], Johanna Schüttler [7], Marcus Sundbom [8,13], Sinhué Torres-Valdés [3], Stefan Bertilsson [9], Ilka Peeken [3], Pauline Snoeijs-Leijonmalm [10] & Lasse Riemann [1] ✉

With climate change-induced sea ice decline in the Arctic Ocean, nitrogen is expected to become an increasingly important determinant of primary productivity. Nitrogen fixation is the conversion of molecular nitrogen to bioavailable ammonium by microorganisms called diazotrophs. Here, we report nitrogen fixation rates, diazotroph composition, and expression under different stages of declining sea ice in the Central Arctic Ocean (multiyear ice, five stations) and the Eurasian Arctic (marginal ice zone, seven stations). Nitrogen fixation in the Central Arctic Ocean was positively correlated with primary production, ranging from $0.4 \pm 0.1$ to $2.5 \pm 0.87$ nmol N L$^{-1}$ d$^{-1}$. Along two transects across the marginal ice zone, nitrogen fixation varied between days and ice regime from below detection up to $5.3 \pm 3.65$ nmol N L$^{-1}$ d$^{-1}$ associated with an ice-edge phytoplankton bloom. We show nitrogen fixation in sea ice-covered waters of the Arctic Ocean and provide insight into present and active non-cyanobacterial diazotrophs in the region.

The Arctic Ocean plays a large role in global climate regulation through its vast bright sea ice cover, impacting heat fluxes via albedo[1]. The Arctic is warming at rates up to four times faster than the global average[2], which has caused major declines in sea ice coverage, age, and thickness[1]. With decreasing multiyear sea ice and northward expansion of the area where ice melts in summer (seasonal ice zone), the biological productivity of the water column below is impacted and often stimulated through altered stratification and increased light and nutrient availability[3,4]. Between 1998 and 2018, pelagic primary production increased by up to 60% at a pan-Arctic scale[5], driven mainly by increases in the marginal seas across the continental shelves of the Eurasian Arctic Ocean[6]. However, predictions of primary production in these waters and the Central Arctic Ocean (CAO) are complicated by the mosaic of processes affecting nutrient availability[7].

Primary productivity in the Arctic Ocean generally experiences nitrogen limitation with nitrogen:phosphorus (N:P) ratios often below the Redfield ratio[8,9]. Processes that increase nitrogen availability in euphotic

waters can, therefore, stimulate primary production and subsequent carbon sequestration via the biological carbon pump. Nitrogen availability in the Arctic Ocean is governed by multiple processes (e.g., riverine discharge, atmospheric deposition, and advection from the Atlantic and Pacific Oceans) and has so far mainly been addressed from physical and chemical oceanographic points of view[10,11]. Recently, however, the influence of microorganisms on nitrogen availability through e.g., ammonia oxidation and nitrification has been acknowledged[12,13]. Nitrogen fixation is the reduction of molecular nitrogen to bioavailable ammonium by a functional group of microorganisms called diazotrophs. It is a biologically mediated process previously not thought to occur in the Arctic Ocean[14]. It has the potential to increase the input of new nitrogen in the Arctic Ocean[15], and thereby support primary production. However, this prediction suffers from data shortage from all Arctic Ocean regions beyond the Amerasian inflow and internal shelves, and the northern Baffin Bay[16–19]. Hence, to predict nutrient availability and future primary production in the rapidly changing Arctic Ocean, there is a need to assess the magnitude and environmental

[1]University of Copenhagen, Department of Biology, Helsingør, Denmark. [2]Linnaeus University, Centre for Ecology and Evolution in Microbial Model Systems (EEMiS), Universitetsplatsen 1, Kalmar, Sweden. [3]Alfred Wegener Institute, Helmholtz Centre for Polar and Marine Research, Am Handelshafen 12, Bremerhaven, Germany. [4]Aix Marseille University, Université de Toulon, CNRS, IRD, Marseille, France. [5]Turing Center for Living Systems, Aix-Marseille University, Marseille, France. [6]National Oceanography Centre, European Way, Southampton, United Kingdom. [7]Max Planck Institute for Chemistry, Hahn-Meitner-Weg 1, Mainz, Germany. [8]Stockholm University, Department of Environmental Science (ACES), Stockholm, Sweden. [9]Swedish University of Agricultural Sciences, Department of Aquatic Sciences and Assessment, Uppsala, Sweden. [10]Stockholm University, Department of Ecology, Environment and Plant Sciences, Stockholm, Sweden. [11]Present address: Linnaeus University, Centre for Ecology and Evolution in Microbial Model Systems (EEMiS), Kalmar, Sweden. [12]Present address: UiT The Arctic University of Norway, Department of Arctic and Marine Biology, Tromsø, Norway. [13]Present address: SCB Statistics Sweden, Solna, Sweden. ✉e-mail: lriemann@bio.ku.dk

regulation of nitrogen fixation, and the distribution, activity, and metabolic function of key diazotrophs.

Cyanobacterial diazotrophs are thought to dominate nitrogen fixation in lower-latitude oceans[20]. In contrast, the sparse data available suggest that diazotrophs are dominated by non-cyanobacterial diazotrophs (NCDs) in the Eurasian inflow, outflow, and more central parts of the Arctic Ocean[18,21–25]. However, NCDs have not yet been directly linked to nitrogen fixation activity in the Arctic. Nevertheless, their dynamic spatiotemporal patterns in distribution, abundance, and activity in the ocean suggest a tight integration with the biogeochemical cycling of nitrogen[26]. NCDs are thought to be hetero- or mixotrophic, thus relying on organic carbon for their metabolic activity[27]. Dissolved organic carbon (DOC) has previously been documented as a limiting factor for nitrogen fixation in marine waters[28,29]. This demonstrates that DOC availability can be an important factor in regulating nitrogen fixation with a putative contribution from NCDs. Moreover, there is growing evidence that particulate organic matter (POM), especially planktonic aggregates, provides favourable conditions for nitrogen fixation by NCDs through the provision of labile organic carbon, anoxic microenvironments, trace metals, and phosphorus[30]. Indeed, POM has been suggested to influence diazotroph community composition in Arctic ecosystems (Young Sound, east Kalaallit Nunaat (Greenland))[23] and to enhance nitrogen fixation in the Mackenzie River estuary (Canadian Arctic)[17]. With ongoing increases in Arctic riverine discharge and changing primary production in the Arctic marginal seas, the quality, quantity, and distribution of organic matter are bound to change, with a yet unknown impact on nitrogen fixation.

Phytoplankton and associated dissolved and particulate organic matter often accumulate at a position in the water column known as the deep chlorophyll $a$ maximum (DCM)[9]. At the DCM, elevated nitrogen fixation has been documented in the Canadian Arctic[17]. In this study, we document nitrogen fixation and diazotroph community composition, abundance, and activity at the DCM and experimentally investigate the potential governing role of DOC in the central and western Eurasian Arctic Ocean (Fig. 1, Figure S1). We show potential links between nitrogen fixation, primary production, NCDs, and organic matter. By sampling contrasting sea ice regimes in the CAO, across the marginal ice zone (MIZ), and under coastal land-fast (LF) sea ice, we assess the magnitude, environmental regulation, and potential importance of nitrogen fixation for new primary production (i.e., the amount of new nitrogen introduced in the system) across the withdrawing Arctic sea ice cover.

## Results

### Hydrography and sea ice regimes

The sampled stations represented contrasting environmental conditions and DCM depths (Fig. 2, Figure S2, and Figure S3; Table S1). In the Central Arctic Ocean (CAO), salinity and temperature at the DCM varied between 30.37 and 32.73 and −1.71 to −1.50 °C, respectively (Table 1; Figure S2) with relatively high concentrations of phosphate ($PO_4^{3-}$; $1.17 \pm 0.23$ μM) and moderate dissolved silicate (dSi; $4.34 \pm 2.76$ μM; Fig. 2; Table 1). Stations 18, 26, and 38 were covered with multiyear sea ice (mainly second-year ice: thickness ≤ 3.2 m). Less compact decaying multiyear ice (thickness: 1–1.2 m) occurred at Station 50 in the Wandel Sea and at Station 56_2021 in the Nansen Basin (Figure S4). Stations 18, 26, and 38 are therefore defined as a separate sea ice regime (multiyear ice) from Stations 50 and 56_2021 (decaying multiyear ice; Table 1).

Across the marginal ice zone (MIZ), the salinity and temperature at the DCM were higher than in the CAO (33.22 to 34.74 and −1.62 to 5.56 °C, respectively; Table 1; Figure S2). The MIZ encompassed the sea ice regimes of Atlantic open water (i.e., no sea ice), high melt (pulse of meltwater from actively melting sea ice), and pack ice (first-year pack ice, ~1.3 m; Table 1; Figure S2). The first MIZ transect was characterised by higher nitrate ($NO_3^-$) concentrations ($5.30 \pm 2.12$ μM) and nitrogen to phosphorus (N:P) (($NO_3^- + NO_2^- + NH_4^+$)/ $PO_4^{3-}$) molar ratios ($11.9 \pm 1.2$; Fig. 2; Table 1). The second MIZ transect had lower $NO_3^-$ concentrations ($1.83. \pm 1.79$ μM) and N:P molar ratios ($5.25 \pm 4.94$) but higher chlorophyll $a$ (up to 9.0 μg L$^{-1}$) and particulate organic carbon (POC) concentrations (up to 63.4 μM; Fig. 2; Table 1). An ice edge phytoplankton bloom developed between the first and second transect (seven days between the transects) with increased primary production in the high melt and pack ice regimes of transect two (Stations 56_2022, 61 and 65; Fig. 3A). Judged by the decrease in dSi (2.76 to 0.12 μM)

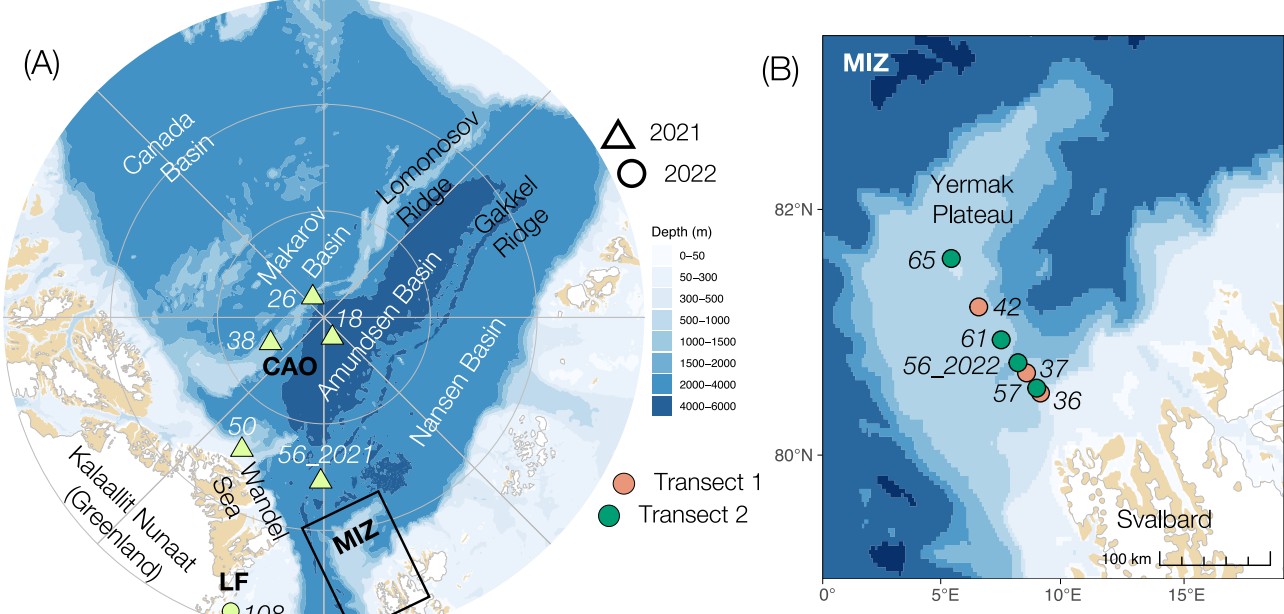

**Fig. 1 | Maps of the study regions. A** North-Pole-centred overview of the Central Arctic Ocean (CAO) and the stations sampled during the Synoptic Arctic Survey (SAS) in 2021. Note that Station 108 in northeast Greenland under land-fast (LF) ice was sampled during ATWAICE in 2022. The Marginal Ice Zone (MIZ) region is marked with a black box. **B** Enlargement of the MIZ region and the stations sampled along two transects during the ATWAICE cruise 2022. Note that two different stations named 56 were sampled in 2021 (**A**) and 2022 (**B**), respectively. The map delineates study areas and does not necessarily depict accepted national boundaries. The bathymetric legend is for (**A**).

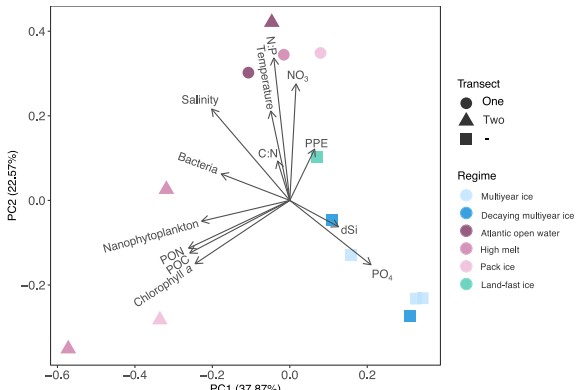

**Fig. 2 | Environmental differences between the study regions.** The two first principal components from principal component analysis (PCA) are displayed (eigenvalues PC1: 4.92, PC2: 2.94). Transects one (circle) and two (triangle) refer to the two transects performed over the marginal ice zone; the square symbol displays all other stations. PON particulate organic nitrogen, POC particulate organic carbon, chlorophyll a (>0.7 μm ATWAICE, >0.3 μm SAS), C:N carbon (POC) to nitrogen (PON) molar ratio, $PO_4^{3-}$ phosphate, dSi dissolved silicate, $NO_3^-$ nitrate + nitrite, N:P nitrogen to phosphorus molar ratio, PPE photosynthetic picoeukaryotes.

(Table 1) and increase in fucoxanthin in the pack ice region (0.26 to 5.71 μg L$^{-1}$; Figure S5; Table S3), the bloom was dominated by diatoms. An increase of 19'-hexanoyloxy-4-ketofucoxanthin (Hex-keto) from 0.06 to 0.34 μg L$^{-1}$ in the high melt regime also suggests an increased abundance of *Phaeocystis* sp. relative to the first transect (Figure S5; Table S3).

Station 108 in the vicinity of the 79°N glacier (northeast Greenland) was considered a separate ice regime due to its cover of land-fast (LF) sea ice ( < 1 m) and proximity to the coastline. This latter region featured low inorganic and organic nutrient concentrations ( < 1 μM) (Table 1), a high carbon to nitrogen (C:N) molar ratio (POC: particulate organic nitrogen (PON) 30.8), and $62 \pm 17\%$ more coloured dissolved organic matter (cDOM) relative to the MIZ stations (Figure S3).

### Nitrogen fixation and its correlation with nutrients and organic matter

Nitrogen fixation was through stable-isotope tracing detected under all sea ice regimes, with the highest rates in the MIZ (Fig. 3B; Supplementary Datasheet S1). In the CAO, nitrogen fixation rates (0.42–2.53 nmol N L$^{-1}$ d$^{-1}$) were above their respective limit of detection (LOD, average 0.46 nmol N L$^{-1}$ d$^{-1}$) (Supplementary Datasheet 1) at all stations. Nitrogen fixation was highest (2.53 $\pm$ 0.87 nmol N L$^{-1}$ d$^{-1}$) at Station 56_2021 under the decaying multiyear ice regime northeast of Greenland and lowest at Station 18 in the multiyear ice regime of the Amundsen basin (Fig. 3B). Across the MIZ on the Yermak plateau, nitrogen fixation rates (up to 5.30 nmol N L$^{-1}$ d$^{-1}$) were above the LOD at four of seven stations (average LOD 2.50 nmol N L$^{-1}$ d$^{-1}$; Fig. 3B; Supplementary Datasheet 1). Along the first transect, rates were detectable only in Atlantic open waters (Station 36). Along the second transect, rates were detected in all regimes and peaked in the high melt and pack ice regimes (Stations 61 and 65, respectively; Fig. 3B). Note that the rates at Station 61 were only above one of the two LODs. The land-fast ice regime (northeast Greenland, glacier vicinity) featured low but detectable nitrogen fixation rates (0.35 $\pm$ 0.03 nmol N L$^{-1}$ d$^{-1}$; LOD 0.21 nmol N L$^{-1}$ d$^{-1}$; Supplementary Datasheet 1). On average across all samples, the atom% of PON after incubation contributed 65.9 $\pm$ 21.2% to the total error, followed by the atom% of PON at start of incubation (20.2 $\pm$ 22.6%), the atom% of dissolved $N_2$ (9.1 $\pm$ 10.2%), PON concentration (4.8 $\pm$ 9.5%), and time (0.01 $\pm$ 0.01%) (Supplementary Datasheet S1).

Nitrogen fixation increased in response to experimental dissolved organic carbon (DOC) amendment (characteristic for Arctic summer waters) in the CAO at Station 26 (from 0.61 $\pm$ 0.06 to 0.95 $\pm$ 0.12 nmol N L$^{-1}$ d$^{-1}$;

Kruskal-Wallis, $n = 6$, $\chi^2 = 3.86$, $p = 0.049$). There appeared to be a similar stimulation at Station 50 in the Wandel Sea, but this was not statistically significant (from 1.82 $\pm$ 0.37 to 3.17 $\pm$ 1.22 nmol N L$^{-1}$ d$^{-1}$; Kruskal-Wallis, $n = 6$, $\chi^2 = 1.19$, $p = 0.28$; Fig. 3B). Potential stimulation was also seen at Station 56_2022 in the MIZ (Fig. 3B). No significant response to DOC amendment was observed at the other stations (from below LOD to 5.39 $\pm$ 4.37 nmol N L$^{-1}$ d$^{-1}$; Kruskal-Wallis, $n = 6$, $p > 0.05$).

Due to the partly differing analytical methods for environmental parameters and light conditions between the two cruises, separate statistical analyses were performed for CAO (including the CAO-influenced Station 50 in the Wandel Sea) and MIZ (Table 1). Nitrogen fixation rates in the CAO correlated positively with primary production (generalised linear model (GLM), $p = 0.004$; Figure S6). It was also positively correlated with photosynthetic picoeukaryotes (PPE) and negatively correlated with ammonium ($NH_4^+$) (GLM, $p_{PPE} = 0.015$, $p_{NH4} = 0.021$). Nitrogen fixation rates across the MIZ correlated negatively with total nitrogen (TN) and $PO_4^{3-}$ (GLM, $p_{TN} = 0.026$, $p_{PO4} = 0.045$).

### Support of primary production by fixed nitrogen

Primary production, measured with stable-isotope tracing, was low under the land-fast (Station 108), multiyear (Stations 18, 26, and 38) and decaying multiyear sea ice regimes (Stations 50 and 56_2021; Fig. 3A, Supplementary Datasheet S2). Primary production was higher in the MIZ north of Svalbard and increased an order of magnitude between the first and second transect (Kruskal-Wallis, $n = 38$, $\chi^2 = 9.97$, $p = 0.002$; Fig. 3A, Supplementary Datasheet S2). The potential contribution of nitrogen fixation to primary production (based on the amount of new nitrogen added to the system and theoretically becoming available for primary producers through leakage and eventual remineralisation of diazotroph cells) was highest in the decaying multiyear ice regime and in the multiyear ice regime (up to 7.5 and 8.6%, respectively; Table 2). In contrast, the average contribution was <1% across the MIZ (Table 2).

### Presence and activity of diazotrophs

Amplification of the diazotroph marker gene *nifH* was successful for all stations except Stations 26 (CAO) and 37 (MIZ), generating a total of 874 amplicon sequence variants (ASVs). Phylogenetic *nifH* Cluster 1 dominated (92.7%), followed by Clusters 3 (6.4%), 4 (0.8%), and 2 (0.1%). NCDs represented the major proportion of *nifH* reads in all ice regimes (Fig. 4A) and were dominated by the groups Gamma-Arctic1 (relative abundance in CAO: 30.5%, MIZ: 0.02%) and Gamma-Arctic2 (CAO: 64.2%, MIZ: 32.6%; Figure S7A). These groups are placed within the *nifH* gene Subcluster 1 G originally defined and reported active in the marginal (Barents Sea shelf) Eurasian Arctic and proposed as key Arctic NCDs (Figure S7B)[25]. Gamma-Arctic1 and Gamma-Arctic2 *nifH* gene copies quantified with quantitative PCR (qPCR) ranged from below detection (both assays) at the MIZ stations north of Svalbard to a maximum of $1.4 \times 10^2$ (Station 18, CAO, Amundsen Basin) and $2.0 \times 10^4$ gene copies L$^{-1}$ seawater in the Wandel Sea (Station 50, CAO-influenced; Fig. 5). Cyanobacterial diazotrophs (Chroococcales, Pleurocapsales, Nostocales, and *Lusitaniella* sp.) were rare, representing only 0.6% of the total amplicon sequence reads. Candidatus *Atelocyanobacterium thalassa* (UCYN-A), until recently thought to be a photoheterotrophic cyanobacterial symbiont with a haptophyte host but now suggested as an early stage of a eukaryotic organelle[31] (the nitroplast), only represented 0.2% of the total reads with a maximum relative abundance of 5.7% in the large size fraction ( > 2 μm) at Station 38 in the CAO (Fig. 4A).

The diazotroph community composition differed between the CAO (including the CAO-influenced Station 50 in the Wandel Sea) and the MIZ (pairwise adonis, $R^2 = 0.06$, $p = 0.001$), with the CAO dominated by Gammaproteobacteria (Fig. 4A). In contrast, the MIZ harboured diazotrophs mainly from within Gamma-, Beta-, and Deltaproteobacteria (Fig. 4A). Alpha-diversity was higher in the CAO than in the MIZ (Shannon index, one-way analysis of variance (ANOVA), F = 12.95, $p < 0.001$; Figure S8). There was no significant difference in diazotroph community composition between size fractions in the CAO (adonis2, $R^2 = 0.09$, $p > 0.05$; Fig. 4A). The

## Table 1 | Overview of environmental conditions at each station

| Station | Cruise | Region | Regime | Date | Transect # | DCM depth (m) | Temperature (°C) | Salinity | Chlorophyll a (µg L⁻¹) | $NO_3^- + NO_2^-$ (µM) | $NH_4^+$ (µM) | dSi (µM) | $PO_4^{3-}$ (µM) | DOC (mg L⁻¹) | DOP (µM) | DON (µM) | TN (µM) | TP (µM) | POC (µM) | PON (µM) | Bacteria (cells ml⁻¹) | PPE (cells ml⁻¹) | Nanophytoplankton (cells ml⁻¹) |
|---|---|---|---|---|---|---|---|---|---|---|---|---|---|---|---|---|---|---|---|---|---|---|---|
| 18 | SAS | CAO | Multiyear ice | 8/13/21 | - | 22 | −1.7 | 32.73 | 0.4 | 1.1 | 0.10 | 0.9 | 0.92 | 1.3 | NA | NA | NA | NA | 3.2 | 0.4 | 4.0E+05 | 1.9E+03 | 4.8E+02 |
| 26 | SAS | CAO | Multiyear ice | 8/20/21 | - | 30 | −1.6 | 31.03 | 0.1 | 2.2 | 0.09 | 6.7 | 1.24 | 1.8 | NA | NA | NA | NA | 3.5 | 0.2 | 3.0E+05 | 1.8E+03 | 5.5E+02 |
| 38 | SAS | CAO | Multiyear ice | 8/29/21 | - | 33 | −1.6 | 30.37 | 0.2 | 1.7 | 0.09 | 6.5 | 1.40 | 1.6 | NA | NA | NA | NA | 3.6 | 0.3 | 3.7E+05 | 6.1E+03 | 3.1E+02 |
| 50 | SAS | CAO-influenced | Decaying multiyear ice | 9/4/21 | - | 28 | −1.5 | 30.73 | 1.1 | 1.2 | 0.06 | 5.8 | 1.35 | 1.5 | NA | NA | NA | NA | 4.0 | 0.3 | 3.9E+05 | 5.7E+03 | 5.8E+02 |
| 56_2021 | SAS | CAO | Decaying multiyear ice | 9/8/21 | - | 15 | −1.7 | 31.73 | 0.3 | 1.1 | 0.05 | 1.9 | 0.93 | 1.3 | NA | NA | NA | NA | 6.4 | 0.4 | 1.0E+06 | 1.3E+04 | 8.0E+02 |
| 36 | ATWAICE | MIZ | Atlantic open water | 7/11/22 | 1 | 20 | 3.2 | 34.49 | 1.7 | 3.6 | NA | 3.6 | 0.33 | NA | 0.00 | 6.2 | 9.9 | 0.31 | 15.7 | 1.8 | 5.8E+05 | 6.8E+03 | 7.9E+03 |
| 37 | ATWAICE | MIZ | High-melt zone | 7/11/22 | 1 | 35 | −1.4 | 34.27 | 1.2 | 7.7 | NA | 3.7 | 0.58 | NA | 0.00 | 6.5 | 14.2 | 0.50 | 12.8 | 1.2 | 6.2E+05 | 1.9E+03 | 4.0E+03 |
| 42 | ATWAICE | MIZ | Pack ice (first year ice) | 7/12/22 | 1 | 20 | −1.6 | 33.89 | 1.0 | 4.6 | NA | 2.8 | 0.40 | NA | 0.04 | 6.5 | 11.2 | 0.44 | 8.1 | 0.7 | 5.5E+05 | 2.0E+04 | 4.1E+02 |
| 57 | ATWAICE | MIZ | Atlantic open water | 7/18/22 | 2 | 8 | 5.6 | 34.74 | 0.6 | 4.0 | NA | 3.3 | 0.38 | NA | 0.12 | 6.8 | 10.8 | 0.50 | 9.1 | 0.8 | 7.0E+05 | 4.0E+03 | 2.3E+03 |
| 56_2022 | ATWAICE | MIZ | High-melt zone | 7/17/22 | 2 | 31 | −1.2 | 34.08 | 9.0 | 0.8 | NA | 3.4 | 0.39 | NA | 0.28 | 12.5 | 13.3 | 0.67 | 63.4 | 6.3 | 8.0E+05 | 1.4E+03 | 2.0E+04 |
| 61 | ATWAICE | MIZ | High-melt zone | 7/18/22 | 2 | 36 | −0.9 | 34.16 | 4.8 | 2.7 | NA | 3.1 | 0.32 | NA | 0.17 | 7.5 | 10.2 | 0.49 | 30.2 | 3.5 | 9.4E+05 | 3.1E+03 | 1.2E+04 |
| 65 | ATWAICE | MIZ | Pack ice (first year ice) | 7/19/22 | 2 | 11 | −1.4 | 33.22 | 8.8 | 0.0 | NA | 0.1 | 0.01 | NA | 0.17 | 5.4 | 5.4 | 0.18 | 46.7 | 4.1 | 6.0E+05 | 3.2E+03 | 1.7E+03 |
| 108 | ATWAICE | LF | Land-fast ice | 8/6/22 | - | 25 | −1.4 | 31.38 | 0.3 | 0.3 | NA | 0.2 | 0.03 | NA | 0.00 | 1.0 | 1.3 | 0.00 | 7.4 | 0.2 | 3.8E+05 | 2.1E+03 | 7.6E+01 |

SAS Synoptic Arctic Survey (2021), ATWAICE (PS131, 2022), CAO Central Arctic Ocean, MIZ marginal ice zone, LF land-fast ice, DCM deep chlorophyll a maximum (DCM), $NO_3^- + NO_2^-$ nitrate + nitrite, $NH_4^+$ ammonium, dSi dissolved silicate, $PO_4^{3-}$ phosphate, DOC dissolved organic carbon, DOP dissolved organic phosphorous, DON dissolved organic nitrogen, TN total nitrogen, TP total phosphorus, POC particulate organic carbon, PON particulate organic nitrogen, PPE photosynthetic picoeukaryotes, NA not analysed.

**Fig. 3 | Primary production and nitrogen fixation rates.** Primary production rates ($^{13}$C-tracing) (**A**) and nitrogen fixation rates (**B**) at each station. Stars denote statistically significant differences ($p < 0.05$, $n = 3$) between no treatment and dissolved organic carbon (DOC) treatment at the respective station. Error bars represent one standard deviation. NA not analysed, ND not detected (below the limit of detection; Supplementary Datasheets S1, S2), CAO Central Arctic Ocean, MIZ marginal ice zone, Tr. 1 transect one, Tr. 2 transect two, MYI multiyear ice, MYI-decay decaying multiyear ice, AW Atlantic open water, HM high melt, Pack pack ice, LF land-fast ice.

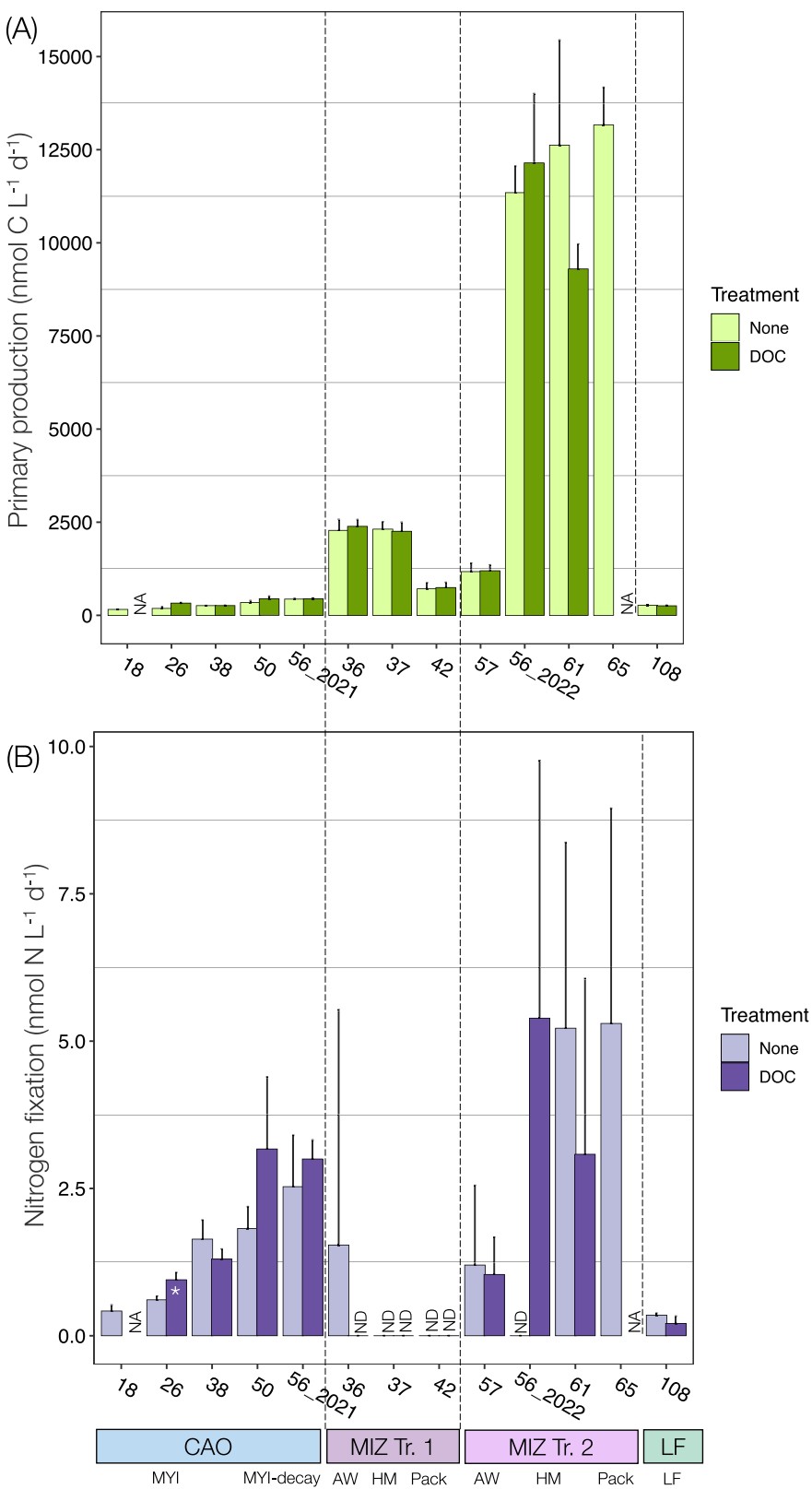

sea ice regimes across the MIZ could not be separated from each other in terms of diazotroph community composition. However, there was an indication of different composition between the two transects (adonis2, $R^2 = 0.31$, $p = 0.07$; Fig. 4A, C). The stations with the highest nitrogen fixation rates across the MIZ (Station 61: high melt, transect two; Station 65: pack ice, transect two) were dominated by Cluster 3 diazotrophs

(subclusters 3T and 3E Desulfobacterales, respectively; Fig. 4A). Under the LF regime, the Gammaproteobacterial genus *Tolumonas* sp. (Subcluster 1E) was highly prevalent (86.0%; Fig. 4A).

The diazotroph community composition in the CAO was correlated with POC, $NH_4^+$, and DOC (redundancy analysis (RDA), F = 1.23, $p = 0.037$; Fig. 4B) and in the MIZ correlated with dissolved organic

**Table 2 | Fraction of primary production supported by nitrogen fixation**

| Station | Region | Regime | Redfield C:N ratio | In situ C:N ratio | |
| --- | --- | --- | --- | --- | --- |
| | | | Fraction of in situ net primary production supported | C:N ratio (molar) | Fraction of in situ net primary production supported |
| 18 | CAO | Multiyear ice | 1.8% | 7.3 | 1.9% |
| 26 | CAO | Multiyear ice | 2.2% | 2.1 | 0.7% |
| 38 | CAO | Multiyear ice | 4.2% | 11.9 | 7.5% |
| 50 | CAO | Decaying multiyear ice | 3.5% | 2.8 | 1.5% |
| 56_2021 | CAO | Decaying multiyear ice | 3.9% | 14.9 | 8.6% |
| 36 | MIZ | Atlantic open water | 0.4% | 8.90 | 0.6% |
| 37 | MIZ | High-melt zone | - | - | - |
| 42 | MIZ | Pack ice (first year ice) | - | - | - |
| 56_2022 | MIZ | High-melt zone | - | - | - |
| 57 | MIZ | Atlantic open water | 0.7% | 11.12 | 1.1% |
| 61 | MIZ | High-melt zone | 0.3% | 8.65 | 0.4% |
| 65 | MIZ | Pack ice (first year ice) | 0.3% | 11.54 | 0.6% |
| 108 | LF | Land-fast ice | 0.9% | 30.84 | 4.0% |

Calculations of the fraction (%) of measured in situ net primary production supported by measured in situ nitrogen fixation rates. Calculations are performed with the Redfield (106:16; Redfield 1934[65]) and station-specific in situ carbon to nitrogen (C:N, i.e., POC:PON) molar ratios, respectively. *CAO* Central Arctic Ocean, *MIZ* marginal ice zone, *LF* land-fast ice. The Stations 37, 42, and 56_2022 do not have values due to nitrogen fixation being below the limit of detection. All limit of detections for nitrogen fixation are found in Supplementary data sheet S1.

phosphorus (DOP), total nitrogen, and fucoxanthin (RDA, $F = 1.84$, $p = 0.0125$; Fig. 4C). In the CAO, *nifH* gene abundance attributed to Gamma-Arctic1 was negatively correlated with cDOM (Spearman, $R = -1$, $p = 0.017$). Gamma-Arctic2 was negatively correlated with phaeophytin ($R = -0.97$, $p = 0.004$) and positively (though not significantly so) with chlorophyll $a > 2\,\mu m$ ($R = 0.82$, $p = 0.09$). The Beta-Arctic1 group was negatively correlated with POC (Spearman, $R = -0.89$, $p = 0.041$).

Amplification of *nifH* genes from cDNA (*nifH* expression) was successful for Stations 18, 50, and 36 and exclusively featured reads from NCDs (Fig. 4A). These were within phylogenetic Clusters 1 and 2 (69.5% and 30.5%, respectively). At Stations 18 (CAO, multiyear ice-covered) and 36 (MIZ, Atlantic open water), *nifH* expression was carried out by Betaproteobacterial ASVs (Subcluster 1 P) of the order Rhodocyclales (Figure S7A). At Station 18, *nifH* genes associated with Subcluster 1 P were additionally quantifiable by qPCR to a maximum of $1.6 \times 10^2$ copies $L^{-1}$ (large size fraction $>2\,\mu m$; Fig. 5). The ASVs showed 98.2–100% *nifH* nucleotide similarity to the Beta-Arctic1 group defined, reported active, and previously proposed as a key Arctic NCD group[25] (Figure S7B). Beta-Arctic1 was present at several stations in our study (Figure S7A) and has previously been reported (*nifH* amplicons from DNA) from Arctic waters[17,18,25]. Firmicutes also expressed *nifH* (subcluster 2A; alternative nitrogenase *anfH*; Fig. 4A) at Station 50 in the CAO-influenced Wandel Sea. Active *nifH* expression from this group has previously been reported in estuaries and terrestrial environments[32]. We did not observe any *nifH* expression of Cluster 3 diazotrophs.

## Discussion

This study expands the known realm of marine nitrogen fixation by reporting nitrogen fixation rates in sea-ice-covered waters of the Central and Western Eurasian Arctic Ocean. Our data suggest that nitrogen fixation in the Arctic is underestimated when sea-ice-covered waters have been excluded, as done earlier[15]. It shows that nitrogen fixation is likely performed by NCDs (i.e., not photoautotrophic), closely tied to primary production and associated metabolic use of dissolved organic matter (DOM) under different sea ice regimes. By covering a range of sea ice regimes, our study assists predictions about future nitrogen fixation in various stages of declining Arctic sea ice under global change.

Sea ice melt may, directly or indirectly, stimulate nitrogen fixation. This is suggested by the detection of the highest nitrogen fixation rates in waters with actively melting sea ice (high-melt and pack ice of the marginal ice zone (MIZ) and decaying multiyear ice of the Central Arctic Ocean (CAO)). This is consistent with findings of elevated nitrogen fixation rates in the vicinity of sea ice compared to open waters in Antarctica[33]. The relationship between sea ice melt and nitrogen fixation may be due to the direct influence of, e.g., iron or DOM release upon sea ice melting and/or indirect effects from ice-edge blooms – a widespread feature of polar oceans. Indeed, nitrogen fixation was positively correlated with primary production in the CAO (including the CAO-influenced Station 50 in the Wandel Sea). In the MIZ north of Svalbard, the peak in nitrogen fixation was associated with the development of an ice-edge bloom. As the identified NCDs are not recognised for being photoautotrophic, the association between nitrogen fixation and primary production is likely indirect, potentially regulated as follows:

First, nitrogen fixation by NCDs is expected to be stimulated by the nutrient concentrations driven by primary producers (i.e., consumed) at the ice edge, where, for instance, diazotrophs conceivably have a competitive advantage when phytoplankton have depleted DIN stocks (e.g., Mills and Arrigo 2010). This agrees with the negative correlation between nitrogen fixation and $NH_4^+$ (CAO) and total nitrogen (MIZ). The negative correlation of nitrogen fixation with $PO_4^{3-}$ in the MIZ may initially seem counterintuitive, given that $PO_4^{3-}$ can limit diazotrophs[34]. However, it may reflect the co-correlation between $NO_3^-$ and $PO_4^{3-}$ ($R^2 = 0.66$ in MIZ; i.e., diazotrophs favoured in dissolved inorganic nitrogen (DIN)-poor conditions) or the utilisation of $PO_4^{3-}$ by diazotrophs themselves. Furthermore, some diazotrophs can utilise DOP[35] which – with an increase from $0.02 \pm 0.02$ to $0.18 \pm 0.07\,\mu M$ between the first and second transect – may suggest that DOP at least partly support diazotroph P-requirements. Similarly, no nitrogen fixation was detected in the LF regime where total phosphorus was below detection. Overall, nutrient availability likely shapes the interaction between primary producers and diazotrophs[36] and, therefore, constrains the potential for nitrogen fixation. Nitrogen fixation may have stimulated primary production along transect two but we judge it as negligible due to the low level of nitrogen fixation compared to the primary production (contributing below 1% of the nitrogen requirement).

Second, increased POC concentrations associated with phytoplankton blooms (e.g., cells, transparent exopolymer particles, faecal pellets, and aggregates) are expected to provide low-oxygen micro-niches as well as accumulation of phosphorus, trace metals and/or labile organic matter with a high C:N ratio favourable for nitrogen fixation by NCDs[30]. Furthermore,

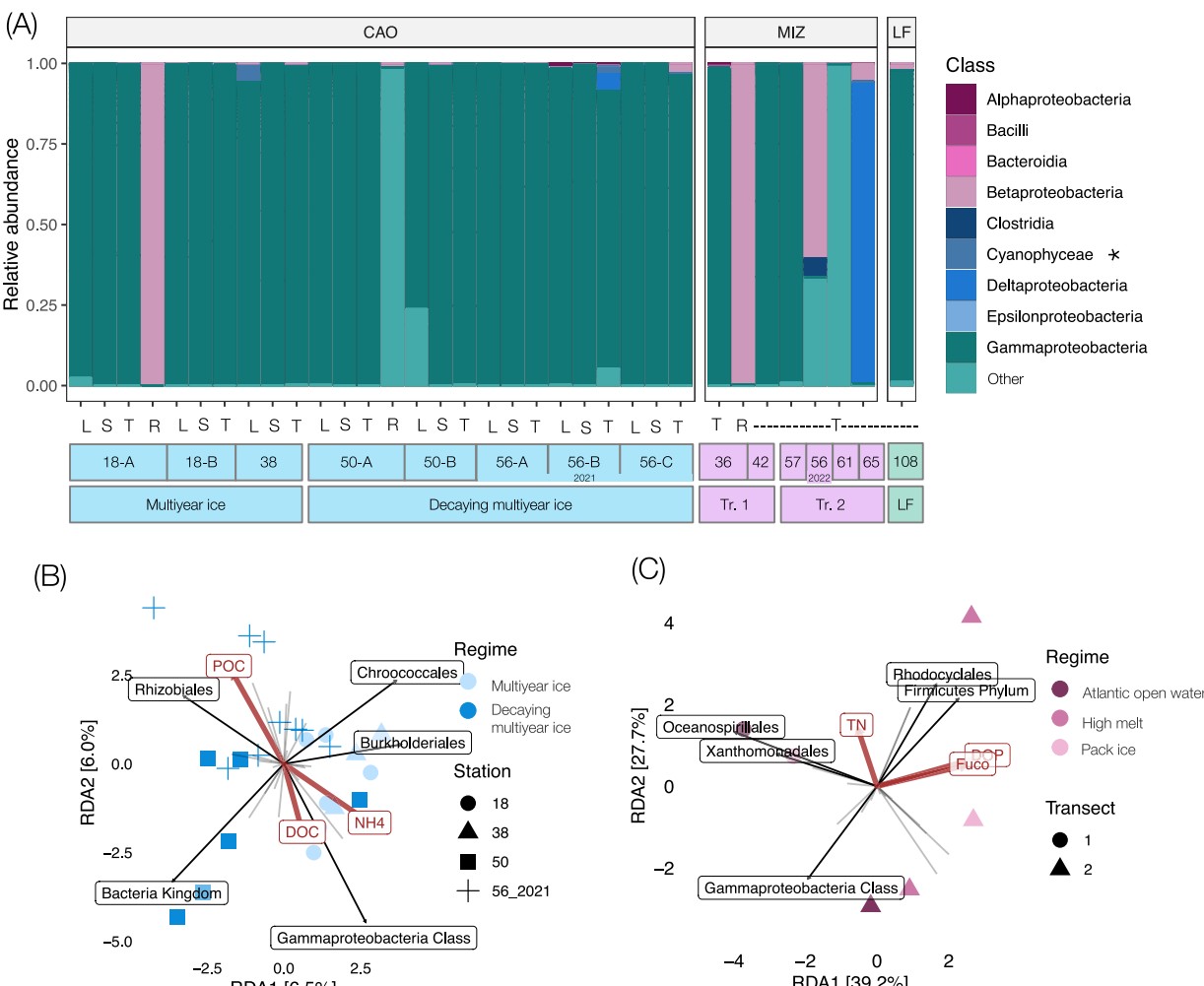

**Fig. 4 | Diazotroph community composition and *nifH* gene expression. A** Relative abundance of diazotroph classes at each station. CAO Central Arctic Ocean, MIZ marginal ice zone, LF: land-fast ice. L: large size fraction ( > 2 μm), S: small size fraction (0.22–2 μm), T total ( > 0.22 μm) size fractions (equal volume of DNA template from >2 and 0.22–2 μm size fractions for comparability of CAO to MIZ and LF, see methods), R RNA (*nifH* expression). Note the one to three replicates (**A–C**) for each station in the CAO. Tr. 1: transect one, Tr. 2: transect two, Other: sequences not annotated at the class level. The asterisk marks the cyanobacterial class category. **B** Redundancy analysis (RDA) of the diazotroph community composition in the CAO, and (**C**) MIZ (transects 1 and 2 refer to the transects performed over the MIZ), displaying the top five orders (if the group does not have assigned taxonomy at order level the closest above-lying available taxonomic rank is given, e.g., Gammaproteobacteria Class) and the significant environmental drivers constraining the RDA (in red). POC particulate organic carbon, DOC dissolved organic carbon, NH₄ ammonium, TN total nitrogen, Fuco fucoxanthin, DOP dissolved organic phosphorus. The light grey lines from the centre in (**B**, **C**) represent the loadings of remaining orders (i.e., beyond the top five displayed orders).

different phytoplankton species will generate variable quantities and qualities of POM and DOM that may, in turn, modulate microbial responses[37]. Notably, the highest nitrogen fixation rates measured during this study were associated with an ice-edge bloom likely dominated by diatoms (pack ice regime) with probable influence from the colony- and mucilage-forming *Phaeocystis* sp. at the southern stations (high melt regime). There are previously described links between NCDs and dSi that initially implicated diatoms as a driver of NCD abundance[25,38]. The NCD GammaA was recently, however, instead proposed as a nitrogen-fixing diatom symbiont[39]. We speculate that such symbiotic relationships may be important in the often diatom-dominated MIZ (both sympagic and pelagic), currently undergoing rapid changes in phytoplankton species assemblage, phenology and productivity[5]. The indicated correlation of Gamma-Arctic2 *nifH* gene abundance to the larger-sized chlorophyll *a* fraction ( > 2 μm) further suggests a potential association of these NCDs with larger phytoplankton such as diatoms. Overall, our findings pinpoint a need for detailed studies on the links between nitrogen fixation by NCDs and specific groups of phytoplankton.

Third, labile DOM leaking from phytoplankton may stimulate mixotrophic and/or heterotrophic diazotrophs[29]. Chemotaxis towards phytoplankton-derived DOM and the widespread capacity for chemotaxis in diverse NCDs[40] support this idea. Excess of phytoplankton-derived DOC could explain the unresponsiveness to DOC amendment across the MIZ, as local primary production levels were high (relative to the CAO, where a response to DOC amendment was observed at Station 26 and indicated at Station 50). The transpolar drift (a major current traversing the CAO east to west) is rich in DOM, but as this is largely of terrigenous origin from Siberian rivers, it is conceivably less labile than freshly produced phytoplankton-derived DOM. Consistent with the transpolar drift route, higher cDOM was measured at Stations 26, 38, and 50, and the positive nitrogen fixation response to DOC amendment at Station 26 (and indicated at Station 50) shows that nitrogen fixation was, at least regionally, limited by unavailable DOC. In the LF regime, there was high cDOM, which, due to the low local primary production, was likely allochthonous and rather refractory terrigenous DOM. Still, nitrogen fixation did not respond to the amendment of labile DOC in the LF regime, possibly due to phosphorus limitation (see

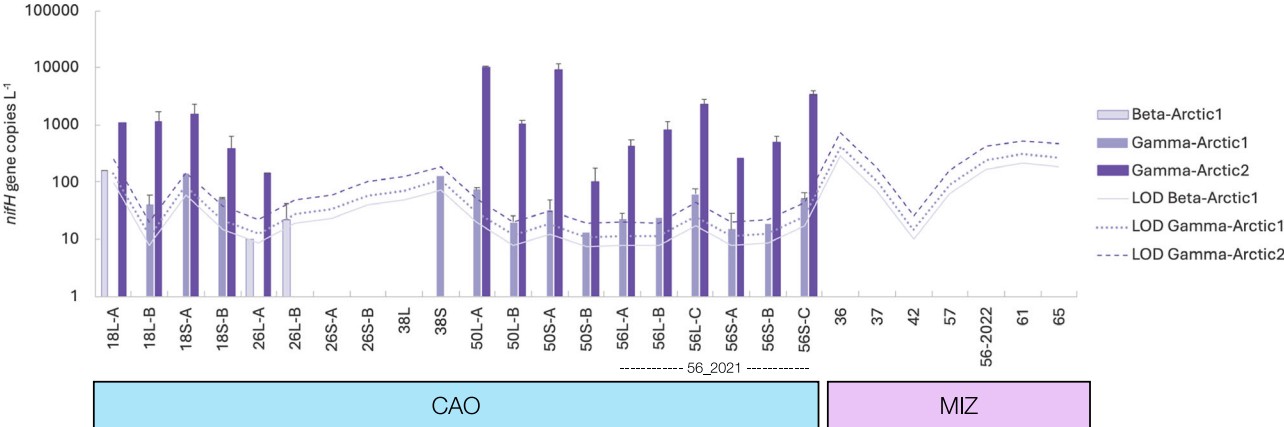

**Fig. 5 | Average *nifH* gene copies of non-cyanobacterial key diazotrophs assessed by quantitative PCR.** Error bars represent the standard deviation for samples with amplification of two or more replicates. Station numbers on the x-axis are followed by the size fraction (L (large >2 μm) or S (small 0.22–2 μm)) and replicate (**A–C**). Only data above the limit of detection (LOD) is presented. qPCR assay efficiencies were 105% (Beta-Arctic1), 110% (Gamma-Arctic1), and 107% (Gamma-Arctic2).

above). Interestingly, *nifH* gene abundances of Gamma-Arctic1 and Gamma-Arctic2 were negatively correlated to cDOM and phaeophytin, respectively, which may indicate their preference for waters with more bioavailable DOM and/or active phytoplankton. Our work suggests that DOM quantity and quality affect nitrogen fixation, which is consistent with findings from, e.g., the North- and South Pacific Ocean and the Mediterranean Sea, where DOC amendments stimulated nitrogen fixation[28,29]. In line with this, higher nitrogen fixation rates have been reported at the DCM than in surface waters in the Canadian Arctic[17]. Given the dominance of NCDs in the Eurasian Arctic Ocean, we speculate that DOM dynamics are particularly decisive for local nitrogen fixation activity in these waters.

Our study supports the notion that NCDs are the main players in nitrogen fixation of the Eurasian Arctic Ocean[21,23]. NCDs dominated *nifH* DNA and cDNA amplicon libraries associated with detectable nitrogen fixation rates, whereas Cyanobacteria were most often absent. In contrast, the coastal Amerasian Arctic hosts active UCYN-A populations[16,19]. Here, we only found UCYN-A1 at Station 38 (above the Lomonosov Ridge between the Amundsen and Makarov basins). This single occurrence may reflect advected Amerasian populations, as was recently suggested for UCYN-A sporadically found in the northern Baffin Bay[24] and across the Barents Sea (there advected from North Atlantic populations[25]). UCYN-A in the Arctic likely consist of both locally sustained[23] and advected populations[18,24], where future study of their ecotype-specific temporal variability[41] would be of considerable interest. However, based on literature values of cell-specific nitrogen fixation rates[42] (2-362 fmol cell$^{-1}$ d$^{-1}$), there would need to be $10^3$–$10^6$ UCYN-A cells L$^{-1}$ to reach the bulk nitrogen fixation rates measured in our samples, a feature which is deemed unlikely to go undetected with the applied nested PCR approach. Thus, we conclude that NCDs are the likely key contributors to the measured nitrogen fixation rates but acknowledge the risk of overlooking some diazotrophs due to primer mismatches[22].

Our data suggest that Beta-Arctic1 is a key NCD group in the Arctic Ocean. Beta-Arctic1 expressed *nifH* under nitrogen-replete conditions (1.1 and 3.6 μM at Stations 18 and 36, respectively) in both the CAO and MIZ, paralleled by nitrogen fixation rates of 0.42 ± 0.1 nmol N L$^{-1}$ d$^{-1}$ and 1.54 ± 3.99 nmol N L$^{-1}$ d$^{-1}$, respectively. Beta-Arctic1 *nifH* genes were also quantifiable at CAO Station 18. We note that *nifH* gene abundances of Beta-Arctic1 were only quantifiable with qPCR in the large size fractions (>2 μm), which may indicate association with a eukaryote host. Beta-Arctic1 showed 98.2% *nifH* nucleotide similarity to *Dechloromonas aromatica* – a chemotactic Betaproteobacterial facultative anaerobe known from soils and freshwaters, potentially associated with a eukaryotic host and capable of degradation of aromatic compounds[43]. Similarly, putative Gammaproteobacterial genera identified in our study are capable of degrading aromatics (in LF: *Tolumonas*[44]; in CAO and MIZ:

Gamma-Arctic2, *Amphritea*[45]). Notably, metagenome-assembled genomes suggest some NCDs can degrade aromatics in the Arctic Ocean[22]. Hence, utilisation of aromatic compounds could be important in Arctic diazotrophs' metabolism, but ecophysiological studies are required to disentangle the significance of this trait.

We propose a ubiquitous distribution of the two NCD groups Gamma-Arctic1 (class unknown) and Gamma-Arctic2 (Oceanospirillales) at the DCM of the CAO. In this study, their *nifH* sequences were widely found in the CAO and (less frequently) in the MIZ – vastly expanding previous observations[21,25]. They were quantifiable with qPCR in the CAO (including the CAO-influenced Station 50 in the Wandel Sea) at *nifH* gene abundances similar to those reported from the Barents Sea[25] ($10^2$–$10^4$ *nifH* copies L$^{-1}$), and at relatively high abundances compared to NCDs in other oceans[27]. Their prevalence in both size fractions in the CAO indicates that Gamma-Arctic1 and Gamma-Arctic2 span both particle-associated (or even symbiotic) and free-living states, but potential filter clogging – an inherent problem in sample size-fractionation – may have contributed to the overlap. Their widespread distribution in the Arctic Ocean contributes to the emerging view that Gammaproteobacteria (and specifically class Oceanospirillales[46]) are key groups of NCDs in the global ocean[22,27,39].

Less prominent ASVs were affiliated with *nifH* clusters 2 and 3. Cluster 2 encompasses alternative nitrogenases[32] (*anfH*: iron-only nitrogenase; *vnfH*: vanadium nitrogenase), here expressed at Station 50 (CAO-influenced Wandel Sea) and represented by Firmicutes *anfH*. This nitrogenase has previously been detected in the Siberian shelf and East Greenland waters[21,47]. The vanadium nitrogenase *vnfH* – being more efficient than *nifH* at lower temperatures[48] – has recently been detected in the MIZ north of Svalbard[25]. These findings warrant further study to understand the distribution and functional significance of alternative nitrogenases in Arctic nitrogen fixation. ASVs affiliating with Cluster 3, which are putative anaerobic and often sulphate-reducing diazotrophs[32], were mainly detected in the high melt and pack ice regimes. Interestingly, Arctic Cluster 3 diazotrophs have been suggested to be of sediment origin[18], and hence, we speculate that they may have been advected with the transport of sediment-laden sea ice[49] across the transpolar drift[21,23].

We propose that the elevated (relative to other stations) nitrogen fixation at stations with actively melting sea ice and higher primary production was caused by the elevated levels of phytoplankton-associated DOC, POM, and/or DOP availability, as well as paralleled by DIN depletion (as also recently modelled for the Amerasian Chukchi shelf sea[50]). If occurring widely, this may generate ephemeral pulses of nitrogen fixation associated with phytoplankton blooms developing along the seasonally retreating ice edge. Lower but potentially more constant background nitrogen fixation rates appear in ice-covered (here: multiyear ice in the CAO) and open waters (here: open Atlantic waters), likely limited by the

availability of labile DOC and possibly also $PO_4^{3-}$. When analysing biomass-independent specific nitrogen fixation rates (Figure S9) in addition to the regular biomass-corrected nitrogen fixation rates, it is visible that the contribution of nitrogen fixation to the total local nitrogen turnover is higher in the CAO than the MIZ. The CAO may thus i) provide a more important niche for Arctic diazotrophs, and ii) be a region where nitrogen fixation is of higher ecological relevance (Figure S9).

Ongoing climate change expands the seasonal ice zone as the sea ice retreats further in summer than in winter[1]. This alters phytoplankton bloom phenology, location, and community composition[5], and can regionally enhance salinity-driven stratification, causing prolonged periods of nitrogen limitation[51] in combination with a strengthened nutricline and DCM[4,9]. Therefore, we hypothesise that there will be local shifts in the magnitude of nitrogen added through nitrogen fixation and consequent potential stimulation of primary production in the future Arctic Ocean. In turn, phytoplankton-derived DOC would likely nourish non-cyanobacterial nitrogen fixation further. Hence, the ultimate disappearance of summer sea ice is likely to alter the biogeochemical cycling of nitrogen with pronounced consequences for new primary productivity and carbon sequestration in the Arctic Ocean. We consequently encourage a future modelling effort targeting the magnitude and dynamics of nitrogen fixation in the Arctic Ocean. This should include nitrogen fixation in sea ice-covered waters and consider the high spatiotemporal variability of nitrogen fixation across the heterogeneous Arctic Ocean.

## Methods
### Study regions
The study is based on two scientific cruises in different regions of the Arctic Ocean. The first cruise was conducted with IB Oden during the Swedish Synoptic Arctic Survey (SAS) expedition (Fig. 1A) from 26 July to 19 September 2021, covering various states of multiyear sea ice (Table S1). This first cruise visited four stations (Stations 18, 26, 38, and 56_2021) in the Central Arctic Ocean (CAO; PAME[52],) and one station (Station 50) in the CAO-influenced Wandel Sea north of Greenland. These five stations we refer to as CAO. The second cruise was conducted with RV Polarstern during the PS131 expedition (ATWAICE) from 28 June to 17 August 2022 at the Yermak Plateau north of Svalbard (Fig. 1B; Table S1). There, a two-times repeated transect for the assessment of temporal patterns across various sea ice regimes (from Atlantic open water to first-year pack ice) in the marginal ice zone (MIZ) was performed. This second cruise visited seven stations in the MIZ (Stations 36, 37, 42, 57, 56_2022, 61, and 65), and additionally, one station (Station 108) was located under land-fast ice in the vicinity of the 79°N glacier in northeast Greenland (Fig. 1A).

### Sample collection and physicochemical measurements
The DCM depth was identified from CTD profiles (SBE911 + , ECO-AFL/FL (SAS), ECO FLRTD (ATWAICE); Sea-Bird Scientific, WA, USA; Table S1) and sampled using Niskin bottles (12 L) mounted on a rosette. CTD profile data are available for both cruises (see Data availability). Calibration of SAS CTD sensors was for fluorescence against chlorophyll $a$ (ethanol extraction, fluorometry[53]), and for cDOM (a marker for water origin and DOM quality in the Arctic) with Raman[54]. Raw values (i.e., relative) from the fluorescence and cDOM sensors were used from ATWAICE. The concentration of dSi during the SAS expedition was determined onboard (Seal Analytical QuAAtro39 autoanalyser, limit of detection (LOD) 0.059 μM), while $PO_4^{3-}$, $NO_3^- + NO_2^-$, and $NH_4^+$ concentrations were measured within 12 months after pre-filtration and freezing at -20 °C using a 4-channel continuous flow analyser (San + + , Skalar, Klaipeda University, Lithuania, LODs: 0.059, 0.083, 0.056, and 0.2 μM, respectively). Uncertainty may be associated with the analysis of $NH_4^+$ in frozen samples, and these data should therefore be interpreted with caution. Nutrient concentrations were determined onboard during ATWAICE ($PO_4^{3-}$, $NO_3^- + NO_2^-$, total N, total P, dSi; Seal Analytical AA-500 autoanalyser)[55]. The LODs were: 0.14, 0.006, 0.02, 0.01, 0.09, and 0.01 μM for $NO_3^- + NO_2^-$, $NO_2^-$, dSi, $PO_4^{3-}$, total N, and total P,

respectively. Dissolved organic nitrogen (DON) and DOP were determined by subtraction of inorganic concentration from the total fraction. DOC was determined using catalytic high-temperature combustion (Shimadzu TOC V analyser) with an acetanilide dilution series as standard[56]. Chlorophyll $a$ was extracted in two size fractions ( > 2.0 μm; 0.3–2.0 μm) and acidified (hydrochloric acid) to obtain phaeophytin concentrations during the SAS expedition[53]. From the ATWAICE expedition, GF/F filters (Whatman, Merck, Germany) for pigment analysis were flash-frozen in liquid nitrogen and stored at −80 °C before analysis with high-performance liquid chromatography as described in Kauko et al.[57] Seawater for cell enumeration using flow cytometry (CYTOFlex, Beckman Coulter Inc, CA, USA; blue laser 488 nm; red laser 638 nm) was fixed (final concentration 0.5% glutaraldehyde, Sigma Aldrich, MA, USA; for SAS samples for eukaryotic cell enumeration including 0.01% Pluronic acid F-68, Gibco, MA, USA), incubated at room temperature for 5 min and frozen at −80 °C. Bacteria were enumerated after staining with SYBR Green I (0.06% final concentration, Invitrogen, MA, USA) with detection of microbial cells using the blue laser (488 nm) and a combination of side scatter and green fluorescence at 525/40 nm. To quantify nanophytoplankton and photosynthetic picoeukaryotes (PPE), 488 nm forward scatter was used as a proxy for cell diameter and red fluorescence at 690/50 nm as a proxy for chlorophyll $a$. *Synechococcus* cells were not detected in any samples. Sea ice thickness and type were determined from nearby sea ice sampling stations (Kovacs corer, thickness gauge measurements) and satellite maps (Norwegian Ice Service).

### Nitrogen fixation and primary production rate measurements
Seawater was collected in acid-washed polycarbonate bottles (2.3 or 4.6 L; Supplementary Datasheet S1), where one to three bottles were filtered immediately (pre-combusted 25 mm GF75, nominal pore size 0.3 μm, Advantec™) to obtain natural $^{15}N/^{14}N$ and $^{13}C/^{12}C$ isotope ratios and amounts of POC and PON. Six bottles were spiked with $NaH^{13}CO_3$ (100 μM; ≥98 atom%, Sigma Aldrich), sealed bubble-free with septa caps, and spiked with $^{15}N_2$ gas (1.3 ml L$^{-1}$; 99%, Cambridge Isotope Labs, MA, USA). Three of the six bottles were also amended with a mixture of semi-labile monosaccharides (no N or P content; 10 μM final concentration) in proportions (in moles, molecular weight of the whole molecules) characteristic for Arctic summer waters (35.1% glucose, 27.1% mannose, 1.7% galacturonic acid, 8.6% rhamnose, 7.5% arabinose[58]) supplemented with 10% pyruvate and 10% acetate. A final bottle served as a control to account for possible naturally occurring changes in the atom% (of PON) during the incubation. This control received only DOC (i.e., no stable isotopes) but was otherwise identical to the rest, and no change in atom% of the particulate organic nitrogen that could influence the results was detected (SAS: $n = 4$, on average an increase of 0.8 ± 0.5%; ATWAICE: n = 7, average a decrease of 0.1 ± 0.4%). Bottles to be incubated were gently shaken for 15 min, and excess gas was removed with a syringe[59]. The bottles were subsequently incubated for 24 h in an on-deck tank with continuously flowing surface seawater (temperature fluctuations during SAS ranged 0.57-2.11 °C, median 0.78 °C; it was not measured during ATWAICE). The tank was covered with neutral-density light-filtration film with a 55% reduction of incident light, a 99% reduction of UV, and a 10% reduction of thermal radiation to simulate in situ conditions. Due to the variable light conditions during ATWAICE (e.g., from open water to pack ice), bottles were additionally shaded by up to three layers of black nylon mesh to resemble light levels at local DCM depths. After incubation, duplicate 12 ml Exetainers (Labco, Lampeter, UK) were siphoned from each bottle to analyse the $^{15}N$ atom% enrichment of the dissolved $N_2$ pool (3.99 ± 1.99%, $n = 142$) with membrane inlet mass spectrometry. Cells and other particles from the remaining water ( ~860–4600 ml) were captured by vacuum filtration onto pre-combusted Advantec™ filters, which were subsequently stored at −20 °C until drying at 50 °C for 24 h and encapsulation in tin cups for analysis with a coupled elemental-analyser to isotope ratio mass spectrometer (Integra 2, SerCon Ltd). Nitrogen fixation rates (nmol L$^{-1}$ d$^{-1}$)[60] and two types of LODs[61] were calculated using the template provided by White et al.[62] (Supplementary Datasheet S1). Biomass-independent specific $N_2$-uptake were additionally

calculated[63] (Supplementary Data sheet S1). Primary production rates were calculated according to Hama et al.[64] (Supplementary Datasheet S2). The contribution of nitrogen fixation to the nitrogen demand for in situ net primary production was estimated using the Redfield ratio[65]. Due to variations in elemental stoichiometry across latitudes[66], the in situ molar C:N ratios were applied in parallel to represent local characteristics.

## Nucleic acid extraction, *nifH* gene amplification, and sequencing

Seawater for collection of nucleic acids was filtered (0.22 µm, Sterivex, Millipore, MA, USA; 30 min; ~1800 ml) in dimmed light, the filter preserved with 1 ml RNAlater (Ambion, CA, USA) and frozen at −80 °C until dual extraction of RNA and DNA (AllPrep DNA/RNA Mini kit, Qiagen Sciences, MD, USA). During SAS, the filtration was done in one to three replicates and size-fractioned via serial filtration through a 2.0 µm filter (large=L; 25 mm, polycarbonate, EMD Millipore, MA, USA) followed by a 0.22 µm filter (small=S, Sterivex). During ATWAICE, water was filtered directly onto a 0.22 µm filter (Sterivex). cDNA was synthesised from RNA (SuperScript IV, Invitrogen, MA, USA) with nifH3 as a gene-specific primer and *nifH* amplicons generated from cDNA and DNA in a nested PCR reaction[67] applying primers nifH1-nifH4 (Table S2) as described earlier[23].

From SAS, DNA of both size fractions were amplified separately (L and S) and additionally pooled by volume (total= T) to obtain data comparable to ATWAICE. Targets were amplified, and sequencing libraries were prepared following von Friesen et al.[23]. Samples from SAS were amplified with MyTaq™ Red DNA polymerase, and the remaining samples were amplified with MyTaq™ HS DNA polymerase (Bioline, Meridian Bioscience, UK) at 95 °C−1 min and 30 cycles of 95 °C−15 s, 50/57 °C (outer/inner)−1 min, 72 °C−1 min, and finally 72 °C-5 min. Two samples only yielded amplification at lower primer annealing temperature (47/54 °C; outer/inner). To control for different amplification protocols, two samples were amplified with all three approaches with no significant differences in the obtained community composition (adonis2, $R^2 = 0.46$, $p = 0.87$; Figure S1). Each successfully amplified RNA sample was tested for DNA contamination by using the equivalent amount of RNA extract as for cDNA synthesis directly in the nested PCR, generating no visible amplification and no more reads than sequenced negative controls (PCR grade UV-irradiated water as a template, see below). Sequencing was performed on an Illumina MiSeqV3 platform (2×300 bp; Geogenetics, University of Copenhagen, Denmark).

## Sequence analysis

Generation, quality control, and taxonomic assignment of ASVs were performed as described in von Friesen et al.[23]. Briefly, we used (1) DADA2[68] (v.1.20.0) for inference of ASVs (trimming: 230 forward, 170 reverse), (2) parts of the NifMAP pipeline[69] to exclude potential *nifH* homologs (v.1.0), (3) assignment of *nifH* phylogenetic clusters[70], and (4) taxonomic assignment with the assignTaxonomy function of DADA2 applying a *nifH* database[71]. After these steps, 878 ASVs were obtained from the 40 environmental and three control samples, with 13,776–199,897 (median 139,579) and 718−1171 reads per sample, respectively. The R-package decontam[72] (v.1.12.0) was applied with the prevalence method (default settings), resulting in the removal of four ASVs. The remaining reads were judged negligible due to the low counts in negative controls: 77 and 430, respectively. The *nifH* gene has been questioned as an explicit marker gene for nitrogen fixation[73]. However, uncertainties concerning pseudo-*nifH* sequences revolve mainly around Clostridia, which only constituted 0.2% of the total relative abundance and were thus considered of minor importance. However, the 5.8% relative abundance of the class Clostridia at Station 56_2022 should be interpreted with caution in terms of their nitrogen fixation potential. Relatively large differences between obtained *nifH* ASVs from DNA and cDNA from the same sample (as seen in our study) are not uncommon and are believed to reflect a situation where only a few diazotrophs are actively transcribing the *nifH* gene at any given moment[74].

## Quantitative PCR

*nifH* gene abundances were quantified for the NCD groups Gamma-Arctic1, Gamma-Arctic2, and Beta-Arctic1 with qPCR using the primer/probe assays developed in von Friesen et al.[25] (Table S2). From SAS, DNA of both size fractions was run separately (L and S) and not pooled together for qPCR. Briefly, the assays were run in two to four replicated 10 µl reactions (2.5 ng DNA per reaction) using a LightCycler® 480 Probes Master (Roche, Basel, Switzerland) with 1x LightCycler® 480 Probes Master mix, 0.5 µM primers, and 0.2 µM probe at 95 °C−300 s, 40 cycles of 95 °C−10 s, 60 °C-30 s, and 72 °C−1 s, and finally 40 °C−10 s. Duplicate ten-fold dilutions ($10^1$–$10^8$) of target standards (gBlocks; Integrated DNA Technologies, IA, USA) were run for each assay. No-template controls (PCR grade water) did not generate any amplification. Inhibition controls (duplicate for each sample) consisted of sample DNA with an added standard concentration of $10^6$ copies $\mu l^{-1}$ and showed negligible inhibition (average 3.5 ± 1.8%). The LOD and limit of quantification (LOQ) were established for each sample and assay[75]. Amplification below LOD was defined as 0 copies $L^{-1}$, and amplification above LOD but below LOQ assigned a conservative number of 1 copy $L^{-1}$. The qPCR data is available in Supplementary Datasheet S3.

## Data analysis

Statistical analyses were performed in R (v.4.1.0), and data were visualised with ggplot2 (v.3.4.0; Wickham[76]) and ggOceanMaps (v.1.3.4; Vihtakari [77]). Differences in nitrogen fixation and primary production rates between sea ice regimes and for evaluation of the DOC treatment were analysed by Kruskal-Wallis tests with Dunn's test for subsequent pairwise comparisons. The potential correlation of nitrogen fixation rates with environmental variables was investigated through z-scoring environmental variables (due to their different units) and evaluated with generalised linear models on $\log_{10}(x + 1)$ transformed nitrogen fixation rates. The relationships between environmental variables (z-scored) and *nifH* gene abundances of specific NCD groups in the CAO were explored through Spearman's rank correlation for monotonic relationships.

Initial pre-processing of *nifH* ASVs in phyloseq (v.1.36.0; McMurdie and Holmes 2013[78]) removed singletons (59 ASVs) and samples with <1600 reads (seven samples, all from Station 26) after inspection of rarefaction curves. The ASV count tables were centered log ratio-transformed for principal component analyses (covariance matrix) using Aitchison distance. The envfit function of the package vegan[79] (v.2.6-2) was used to explore relationships between environmental variables and principal components. Redundancy analysis evaluated the proportion of variation explained by a selection of significant variables from a priori hypotheses and the envfit output. The final RDA model was validated with the functions anova.cca and vif.cca ( < 10) and visualised with the top five influencing orders with microViz[80] (v.0.10.0). Permutational multivariate analysis of variance with adonis2 on Aitchison distance matrices evaluated categorical differences. To ensure homogeneity of group dispersions, the permutest function of the betadisper function output was used in parallel to adonis2. PairwiseAdonis (v.0.4; Martinez Arbizu [81]) evaluated pairwise differences between multiple groups. Bonferroni corrections were used for any multiple comparisons, and 999 permutations were applied for permutational analyses.

## Reporting summary

Further information on research design is available in the Nature Portfolio Reporting Summary linked to this article.

## Data availability

The data that supports the findings of this study are available in the supplementary material of this article, openly available in PANGAEA at https://doi.org/10.1594/PANGAEA.951266 and https://doi.org/10.1594/PANGAEA.956136, and in the National Centre for Biotechnology Information (NCBI) Sequence Read Archive (SRA), reference number PRJNA995422. Supplementary data files 1, 2, and 3, and generated *nifH* amplicon sequence variants are available on figshare: https://doi.org/10.6084/m9.figshare.29930714.

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

## Acknowledgements

We acknowledge the support of the captains and crew from IB Oden (Synoptic Arctic Survey, 2021) and RV Polarstern (PS131, ATWAICE, 2022). The Swedish Polar Research Secretariat (SPRS; https://polar.se) organised and supported the SAS-Oden 2021 expedition with IB Oden. This expedition was the Swedish contribution to the International Synoptic Arctic Survey (SAS; https://synopticarcticsurvey.w.uib.no/). We thank Cecilie Appeldorff (University of Copenhagen), Laura Bas Conn (Linnaeus University), and Erika Allhusen and Sandra Murawski (Alfred Wegener Institute) for laboratory assistance. This work was supported by the Department of Biology, University of Copenhagen [Elite PhD scholarship] to LvF, Independent Research Fund Denmark [6108-00013 and 2032-00001B] to LR, BNP Paribas Foundation [Climate & Biodiversity project NOTION] to MB and LR, the Crafoord Foundation [2020-0881] to HF, the Swedish Research Council [VR, 2018-04685] to PSL, the Swedish Research Council for Sustainable Development [FORMAS, 2018-00509] to PSL, the BIOPOLE National Capability

Multicentre Round 2 funding from the Natural Environment Research Council (NE/W004933/1) to MB, the Swedish Polar Research Secretariat [Synoptic Arctic Survey 2021, Implementation Agreement Dnr 2020−119], the PoF IV program providing ship time on RV Polarstern [Changing Earth – Sustaining our Future, Topic 6.1 of the Helmholtz Association, AWI_PS131_05, AWI_PS131_06, AWI_PS131_07], and the Alfred Wegener Institute Nutrient Facility.

## Author contributions

LWvF, HF and LR designed the study. LWvF, HF, JS and CPL collected the samples. MS and STV analysed nutrients. LWvF and OG performed IRMS and MIMS analyses. MB helped design $^{15}N_2$ incubation experiments. LWvF performed the molecular laboratory analyses, and CPL performed the flow cytometry analyses. WJvA assisted with the interpretation of hydrography and the definition of sea ice regimes. LWvF executed the data analysis and drafted the manuscript with LR. PSL, IP and SB obtained funding for the cruises and supervised the study. All authors edited the manuscript and approved the final version.

## Competing interests

The authors declare no competing interests.
