## [Transparent Peer Review file · Communications Earth & Environment]

Nitrogen fixation under declining Arctic sea ice

Corresponding Author: Professor Lasse Riemann

Version 0:

Decision Letter:

Dear Professor Riemann,

Your manuscript titled "Nitrogen fixation is substantial under declining Arctic sea ice" has now been seen by 3 reviewers, and we include their comments at the end of this message. They find your work of interest, but some important points are raised. We are interested in the possibility of publishing your study in Communications Earth & Environment, but would like to consider your responses to these concerns and assess a revised manuscript before we make a final decision on publication. More specifically, the revised manuscript should moderate its assertions regarding the role of N₂ fixation in supporting primary productivity in the Arctic, compute the propagated error linked to the rates (for both ¹⁵N and ¹³C), and more effectively substantiate measured N₂ fixation rates with microbial ecology data. Additionally, the findings should be placed within a broader context by updating the estimated contribution of the Arctic Ocean to global fixed nitrogen inputs.

We therefore invite you to revise and resubmit your manuscript, along with a point-by-point response that takes into account the points raised. Please highlight all changes in the manuscript text file.

Please submit your point-by-point responses as a separate file, distinct from your cover letter where you can add responses to the Editors' comments that you do not want to be made available to the reviewers. Word files are preferred. We recommend that any figures, tables or graphs that are included in the response to reviewers are also included in the main article or Supplementary Information.

Please use the following link to submit your revised manuscript, point-by-point response to the referees' comments (which should be in a separate document to any cover letter), a tracked-changes version of the manuscript (as a PDF file) and the completed checklist:

Link Redacted

We hope to receive your revised paper within six weeks; please let us know if you aren't able to submit it within this time so that we can discuss how best to proceed. If we don't hear from you, and the revision process takes significantly longer, we may close your file. In this event, we will still be happy to reconsider your paper at a later date, as long as nothing similar has been accepted for publication at Communications Earth & Environment or published elsewhere in the meantime.

Please do not hesitate to contact us if you have any questions or would like to discuss these revisions further. We look forward to seeing the revised manuscript and thank you for the opportunity to review your work.

Best regards,

Annie Bourbonnais, PhD
Editorial Board Member

Communications Earth & Environment
orcid.org/0000-0001-7247-5230

Alice Drinkwater, PhD
Associate Editor
Communications Earth & Environment
Consulting Editor
Communications Sustainability

EDITORIAL POLICIES AND FORMATTING

- Behavioural and social science
- Ecological, evolutionary & environmental sciences
- Life sciences

Furthermore, please align your manuscript with our format requirements, which are summarized on the following checklist: <https://www.nature.com/documents/commsj-phys-style-formatting-checklist-article.pdf> Communications Earth & Environment formatting checklist

and also in our style and formatting guide <https://www.nature.com/documents/commsj-phys-style-formatting-guide-accept.pdf> Communications Earth & Environment formatting guide .

*** DATA: Communications Earth & Environment endorses the principles of the Enabling FAIR data project (<http://www.copdess.org/enabling-fair-data-project/>). We ask authors to make the data that support their conclusions available in permanent, publically accessible data repositories. (Please contact the editor if you are unable to make your data available).

All Communications Earth & Environment manuscripts must include a section titled "Data Availability" at the end of the Methods section or main text (if no Methods). More information on this policy, is available at <http://www.nature.com/authors/policies/data/data-availability-statements-data-citations.pdf>.

If a community resource is unavailable, data can be submitted to generalist repositories such as <https://figshare.com/> or <http://datadryad.org/> Dryad Digital Repository. Please provide a unique identifier for the data (for example a DOI or a permanent URL) in the data availability statement, if possible. If the repository does not provide identifiers, we encourage authors to supply the search terms that will return the data. For data that have been obtained from publically available sources, please provide a URL and the specific data product name in the data availability statement. Data with a DOI should be further cited in the methods reference section.

REVIEWER COMMENTS:

Reviewer #1 (Remarks to the Author):

See attachment.

Reviewer #2 (Remarks to the Author):

Summary

The authors report a study of diazotroph community composition and activity in the Arctic Ocean. Rates and studies of diazotrophy in the Arctic are scarce, so this report is valuable. The authors are well-qualified to carry out the work, the manuscript is well written, clearly organized, and the data are presented effectively. I believe the methods the authors used are robust. The authors speculatively make the case that as oceans warm and stratify it is essential to document the importance of N₂ fixation as a source of N to rapidly changing ecosystems such as the Arctic Ocean. Mostly the authors find low abundances and rates of N₂ fixation in the Arctic. However, the authors frame this result differently, e.g., in the abstract they claim N₂ fixation is "substantial" in the Arctic. I find this framing problematic given the low abundances and low rates of N₂ fixation that they measure. The field of marine N₂ fixation has suffered for decades because of reports of "significant" N₂ fixation when really, rates are low and are not put in a quantitative context. The field now has a credibility problem, and I'm afraid this manuscript will further exacerbate the problem. While it's useful to document diazotrophy robustly, and I believe the authors have been careful with their methods, which I very much appreciate, it's still critical that the quantitative significance of N₂ fixation in an environment is placed in the appropriate, quantitative context. This includes adjusting statements that make it sound like 10% of NPP was supported across the Arctic by N₂ fixation, when it really was at a minor fraction of the stations where not even 10% of NPP was supported by N₂ fixation (see Table 2). It's also difficult to support the prediction that N₂ fixation rates will increase in the Arctic as it warms; here, the authors speculate.

Major Comments

-I would like to see the authors do an analysis similar to that in Turk-Kubo et al., 2013, Environmental Microbiology, doi:10.1111/1462-2920.12346, where they compare N₂ fixation rates estimated using ¹⁵N₂ uptake with diazotroph abundance estimated using similar nifH techniques described here, and literature-based estimates of per-cell N₂ fixation rates. Indeed, the authors reference this calculation, so I was surprised that they didn't do a similar calculation with their data. Showing that literature-based per-cell N₂ fixation rates for the diazotrophs they identify using molecular tools multiplied by the diazotroph abundances they estimate are capable of generating the ¹⁵N₂-based N₂ fixation rates they estimate would strengthen this study.

Minor Comments

-lines 175-177: total N and total P are not inorganic nutrient concentration analyses; please revise

-lines 353-355: Please also report the NO₃+NO₂ and NH₄ concentrations at the same depth

-line 367: Please clarify – if the N:P molar ratio you're reporting is water column inorganic nutrient concentration ratios, can you specify that? If it's something else (e.g., biomass or particulate stoichiometry? DON:DOP concentration ratios?) please indicate that; same on line 369

-line 383: I'm not sure what C:N ratio you're reporting here – if it's POC:PON, please specify. If it's other, please specify that, as well; same for Fig. 2 and Table 2 captions

-DON and DOP concentrations are reported in Table 1, but no methods are included for those analyses – please include methods for these measurements

-lines 421-424: Can you include the un-amended as well as DOC amended N₂ fix rates for both Stn2 26 and 56 described here so the reader can get a sense for the magnitude of change in the rates?

-line 569: recommend changing "substantial" to "detectable"

Reviewer #3 (Remarks to the Author):

Summary This manuscript looks at N₂ fixation in the Arctic, a region where rates are low and variable. The manuscript is well-presented with clear figures and easy to follow text. I believe the lead author is a PhD student and they should be commended for assembling this well-formulated manuscript. My major and minor comments are listed below and they should be answered prior to publication. Well done on this nice piece of work.

Major comments.

1) There is a big difference between the level of confidence displayed in the abstract compared to the level of confidence displayed in the conclusion about the results and their analysis. The discussion presented a more balanced consideration which was missing in the abstract. I encourage the authors to read Gu (2025) Evidence of limited N₂ fixation in the Southern Ocean. *Comm Earth & Env* 6: 264 which presents a balanced overview on the potential importance of N₂ fixation in polar environments.

2) It is well-known that rate measurements of $^{15}\text{N}_2$ assimilation have high variability, and can approach 20% variability. This makes the measurements challenging in the oceans where N_2 fixation is more prolific (5-10 $\text{nmol L}^{-1} \text{d}^{-1}$). For marine environments where rates are close to detection limit, I think greater emphasis needs to be placed on improving and accuracy. This study uses deckboard incubations of three samples with no report of temperature levels during the incubation. The authors also rely on ^{13}C -assimilation for estimating net primary production which is known to be less accurate than O_2 -based methodologies (when conducted correctly). I would like to see Table 2 include the error propagated from both the ^{13}C and ^{15}N rate measurements, which will highlight the uncertainty associated with N_2 fixation in the Arctic.

Minor comments

Abstract.

Sentence 1 nitrogen already is an important determinant of primary productivity

Sentence 4 After you have used the chemical formula (N_2) in sentence 2 you need to use it thereafter

Sentence 4 Data shortage impedes understanding nitrogen fixation supported productivity in the global oceans, not just Arctic

Is the positive correlation between N_2 fixation and primary production an auto-correlation? If yes, then should it be highlighted in the abstract as a key finding?

I struggle with the summaries of N_2 fixation and productivity/phytoplankton in the Central Arctic Ocean and Eurasian Arctic. For one site you mention rates of net primary production while at the other site you mention a phytoplankton bloom but no rate of net primary production. Also if one site ranges from 0.4 to 2.5 $\text{nmol L}^{-1} \text{d}^{-1}$ and the other site ranges from ND to 5 $\text{nmol N L}^{-1} \text{d}^{-1}$, what is the difference?

Introduction

Line 61 – what do you mean by altered stratification? This is quite vague and it would be helpful to know if you mean weakened or strengthened

Line 83 I am very wary of over-exaggerating the importance of N_2 fixation in the Arctic.

Method

Nice charts/locations

Line 163 - Why did you target the DCM? What depths did other N_2 fixation studies target?

Line 206 – Why would you use ^{13}C for measuring net primary production? You cite Hama (1993) as the method, but that's 30 years old and methods have improved. The issue with ^{13}C is its accuracy. There are much more reliable measurements using O_2 -based methodologies.

Line 208 – Triplicate measurements are the minimum for replication and when there is high uncertainty maybe increasing this to 4 or 5 would be helpful?

Why don't you deploy a sediment trap and look at the $\delta^{15}\text{N}$ signal of sinking material?

Line 220 – Did you put a temperature logger in the on-deck tank? Its very easy for on-deck incubators to end up being a few $^{\circ}\text{C}$ higher than the ambient water-column and I have seen a few case studies were it was 3-5 $^{\circ}\text{C}$ higher!

Line 235 – Thanks for providing data

I don't have comments about the sequencing methods

Results

Figure 2 I don't understand this figure. Are you suggesting that the silicate and phosphate in multi-year ice is a driving factor? Why then did you get the phytoplankton bloom in the transect

Does Figure 2 repeat information shown in Table 1?

Line 405 'Nitrogen fixation was detected under all sea ice regimes' doesn't this contradict what you stated in the abstract

Line 421 Does stimulation within 24 hours surprise you? How is the community able to respond so quickly? Does this reflect an increase in growth rates of bacteria or is nitrogenase activated in the wider microbial community

Figure 3A Why does DOC decrease primary production at Station 61? Please can this graph be labelled ^{13}C -primary production

Figure 3B The $\leq 20\%$ variability in rate measurements is always tough (just an observation).
Discussion

The discussion is very long (>2400 words). You have 4 pages linking N2 fixation to primary production alone.

What is the take-home message about NCDs. More variation in the MIZ – why is this?

Line 607 Transect line or line of thinking

Line 608-614 Thank you!

Supplementary Info

I appreciated the transparency of data

Communications Earth & Environment is committed to improving transparency in authorship. As part of our efforts in this direction, we are now requesting that all authors identified as 'corresponding author' create and link their Open Researcher and Contributor Identifier (ORCID) with their account on the Manuscript Tracking System prior to acceptance. ORCID helps the scientific community achieve unambiguous attribution of all scholarly contributions. You can create and link your ORCID from the home page of the Manuscript Tracking System by clicking on 'Modify my Springer Nature account' and following the instructions in the link below. Please also inform all co-authors that they can add their ORCIDs to their accounts and that they must do so prior to acceptance.

Version 1:

Decision Letter:

Dear Professor Riemann,

Your manuscript titled "Nitrogen fixation under declining Arctic sea ice" has now been seen by our reviewers, whose comments appear below. In light of their advice we are delighted to say that we are happy, in principle, to publish a suitably revised version in Communications Earth & Environment.

We therefore invite you to revise your paper one last time to address the remaining concerns of our reviewers. At the same time we ask that you edit your manuscript to comply with our format requirements and to maximise the accessibility and therefore the impact of your work.

EDITORIAL REQUESTS:

*****Please take care to match our formatting and policy requirements. We will check revised manuscript and return manuscripts that do not comply. Such requests will lead to delays. *****

SUBMISSION INFORMATION:

OPEN ACCESS:

Communications Earth & Environment is a fully open access journal. Articles are made freely accessible on publication. For further information about article processing charges, open access funding, and advice and support from Nature Portfolio, please visit <https://www.nature.com/commsenv/open-access>

Link Redacted

Best regards,

Alice Drinkwater, PhD
Associate Editor
Communications Earth & Environment
Consulting Editor
Communications Sustainability

REVIEWERS' COMMENTS:

Reviewer #1 (Remarks to the Author):

I am satisfied that the authors have addressed my comments, and I appreciated their updated N₂ fixation calculations. I have only a few minor comments (associated with line numbers from the revised, marked up text):

[abstract] In line with comments from the other reviewers, I suggest adding some indication of error magnitude where Nfix rates are reported in the abstract. It helps contextualize the numbers for those who are interested in using them, but may not fully understand where they come from.

[217] It would be useful for a casual reader here to specify why two LOD calculations were made and what the difference is.

[222-224] I would point out to the authors that error propagation is standard practice across analytical chemistry and the template provided by White et al simply applies this practice in the context of N₂ fixation rates. In other words, there is nothing that need be "developed". That said, I think it is okay not to do this for Cfix (as suggested by reviewer 3) because the authors already report standard deviation across replicates, which includes both analytical and biological variability in the measurement. Moreover, unlike with N₂ fixation, ¹³C-Cfixation rates generally have signals that are greater than analytical noise and thus easier to distinguish without analyzing sources of analytical error.

Reviewer #2 (Remarks to the Author):

I am mostly satisfied with the responses from the authors, thank you.

Lines 157-161: However, the authors still have not provided the methods they used to measure TDN and TDP concentration. Was TDN measured with DOC by high temperature combustion? Was persulfate oxidation or UV oxidation used? Similarly, for TDP, was the ash-hydrolysis method used? UV oxidation? Please specify the method used and provide references for the method, as well as precision of the method (i.e., +/- 0.6 μM TDN). Same for inorganic nutrient concentration measurements.

Reviewer #3 (Remarks to the Author):

The authors have responded to the questions raised and I do not think the manuscript can be substantially improved. I recommend publication. Good work!

** Visit Nature Portfolio's author and referees' website at www.nature.com/authors for information about policies, services and author benefits**

To the editors

Dear Associate Editor Alice Drinkwater and Editorial Board Member Annie Bourbonnais of Communications Earth & Environment,

Thank you for facilitating the peer review of our manuscript “Nitrogen fixation is substantial under declining Arctic sea ice” (COMMSENV-25-1726-T). We have carefully gone through and responded to each comment provided by the three reviewers. We are pleased to now submit a thoroughly revised version of the manuscript. Our responses are written in italics, and all changes can be traced through tracked changes in one of the manuscript files. The line numbers in our responses refer to the manuscript with visible tracked changes.

We want to thank the reviewers for their qualified, thorough, and constructive comments.

Best regards,

Lisa W. von Friesen and Lasse Riemann, on behalf of all authors

Editorial comments

Comments to the Author

Your manuscript titled "Nitrogen fixation is substantial under declining Arctic sea ice" has now been seen by 3 reviewers, and we include their comments at the end of this message. They find your work of interest, but some important points are raised. We are interested in the possibility of publishing your study in Communications Earth & Environment, but would like to consider your responses to these concerns and assess a revised manuscript before we make a final decision on publication. More specifically, the revised manuscript should moderate its assertions regarding the role of N₂ fixation in supporting primary productivity in the Arctic, compute the propagated error linked to the rates (for both ¹⁵N and ¹³C), and more effectively substantiate measured N₂ fixation rates with microbial ecology data. Additionally, the findings should be placed within a broader context by updating the estimated contribution of the Arctic Ocean to global fixed nitrogen inputs.

Response:

We are happy to see that all three reviewers support our study. We have carefully revised the manuscript, with specific emphasis on the following:

- Adjusted the phrasing of the importance of Arctic nitrogen fixation to better reflect the underlying data and specific environment. E.g., we have revised the title, rephrased the abstract, and throughout the manuscript been more careful with suggesting the importance of nitrogen fixation.*
- More clearly, presenting error-propagation calculations of nitrogen fixation rates, as well as providing biomass-independent specific nitrogen fixation rates (new figure)*

Further, we provide our response to the suggestion on providing an updated model of the global importance of Arctic nitrogen fixation. The key message from our data is the reporting of nitrogen fixation rates under sea ice, but also of a high spatiotemporal variability in nitrogen fixation magnitude between the areas sampled. We argue that an updated solid pan-Arctic estimation of nitrogen fixation is currently not feasible. Moreover, the generation of such a budget, as requested by Reviewer 1, would go against the request of Reviewer 2 to not over-interpret the importance of Arctic nitrogen fixation.

Reviewer 1

Summary

This study reports N₂ fixation rates and diazotroph abundances in the under-sampled Arctic Ocean, with a focus on the relationship of polar diazotrophic to sea ice. N₂ fixation rates were elevated in waters influenced by sea ice melt, which the authors argue is likely due to nutritional coupling between heterotrophic diazotrophs and primary producers.

Response: Thank you for your thorough and constructive review. We have revised the manuscript according to your comments. We believe your review has helped us improve the manuscript.

Comments on quality/impact:

N₂ fixation remains a notoriously difficult process to model and, consequently, represents a critical area of uncertainty in Earth Systems Models as marine primary production is largely nitrogen-limited and sets the upper limit on the biological carbon pump. There is some burgeoning consensus within the field at present that one of the primary factors causing this difficulty is the ecophysiological diversity of diazotrophs. I believe that this study offers two important contributions to the field: (1) It offers new data in a critically under-sampled and historically overlooked region, and (2) it begins to draw a potential mechanistic connection between the activity of presumably heterotrophic polar diazotrophs and productive waters in the marginal ice zone.

The methods applied are robust and the study is appropriately positioned within the context of pertinent literature. Some very minor editing could improve the clarity of the text. (I've tried to help by pointing out a few examples of what I mean in my specific comments below.)

Response: Thank you for your comments, we are happy to hear that you believe the study provides important contributions to the field.

Major comments

The impact of this article could be significantly increased by updating Sipler et al.'s (2017) estimated contribution of the Arctic Ocean to global fixed N inputs with the data reported herein as well as in other recent publications.

Response: We agree with the reviewer that a compilation of available data on nitrogen fixation in the Arctic to conduct a pan-Arctic analysis and identify overarching patterns would be beneficial for the field. However, we believe that this should be done in a targeted study to encompass the spatiotemporal complexity of the process across the heterogeneous Arctic environments (e.g., integration with different sea ice systems, coastal vs off-shore habitats, chlorophyll *a* and inorganic nutrient availabilities, etc) to be useful to the field. We believe that our study provides important aspects to be considered for such a pan-Arctic analysis. In the present study, we focus specifically on nitrogen fixation in connection with sea ice. The main messages of our findings are that:

- i) Nitrogen fixation occurs below sea ice (previously assumed not to be the case in the calculations by Sipler et al 2017, please refer to their Figure 1 pasted here below, where the addition of our data suggests even the calculation of "high open water" is underestimated). This suggests that the previous calculations are

underestimates (as addressed in lines 494-495), but please also see the points below (ii and iii).

Figure 1 from Sipler et al., 2017, doi: 10.1002/lol2.10046

- ii) *Nitrogen fixation is linked to primary production, likely driven by an interplay of inorganic nutrient consumption by phytoplankton and dissolved organic matter dynamics. These findings show that nitrogen fixation is a process that varies in time and space in the heterogeneous environments of the Arctic Ocean. The higher rates reported may be transient spikes of activity associated with, e.g., as shown here, a marginal sea ice bloom. To extrapolate such data over time and space would require consideration of different sea ice systems (marginal ice regime vs pack ice regime), in addition to classification into only open or sea ice-covered systems. Additionally, we recommend integration with available nutrient and chlorophyll data sets for improved accuracy of such calculations. As of today, no similar nitrogen fixation data is available from other sea ice regions in the Arctic (e.g. the Pacific Arctic sector or the inner shelf systems), thus preventing such calculations from taking place without extensive extrapolating assumptions.*

Overall, we find that the current lack of data on nitrogen fixation across the Arctic (few data points, poor geographical and vertical coverage, no information on seasonal variability) makes a pan-Arctic budget highly speculative. We fully acknowledge the high value of Sipler's 2017 study in sparking interest in nitrogen fixation in the Arctic by highlighting the potential biogeochemical ramifications of this process; however, while our study provides critical and novel information about nitrogen fixation under sea ice – and in this way adds data to Sipler's study, we do not find that enough data are yet available to refine the pioneering budget provided by Sipler.

- iii) *Nitrogen fixation is variable in time. With only a week apart, the two transects across the marginal ice zone varied by one order of magnitude. These results complexify the view of Arctic nitrogen fixation, showing high temporal variation in the magnitude of nitrogen fixation rates.*

Conclusively, we do agree with the reviewer that a pan-Arctic compilation of nitrogen fixation would be interesting. However, as stated above, we believe that this should be done in a targeted study to capture enough complexity of the process to be useful to the field. We do believe our manuscript provides essential aspects to be considered in such an updated modelling effort. We have added those aspects and an encouragement and our recommendations for a pan-Arctic compilation in lines 683-687.

The authors stress the potential importance of sea ice melt and ice margin primary producers in stimulating diazotrophy. It's important to note, of course, that mathematically N₂ fixation rates scale with PON, and PON generally scales with primary production. It is thus not very surprising for somewhat higher N₂ fixation rates where PON is higher. We see this in coastal waters where higher Nfix rates occur in high productivity waters even though Nfixers likely make up a comparatively smaller portion of total community production (eg Selden et al 2024, link below). An additional point of interest here is that the relative contribution of N₂ fixation rates (taken either as shown in Table 2 or by looking at specific N₂ uptake rates as in article linked below) is elevated in CAO, basically nil in MIZ Tr. 1, and low-moderate in MIZ Tr. 2. One could interpret this as suggesting a greater niche for diazotrophs in the CAO relative to the MIZ, given that they are contributing more substantially to N turnover under the local conditions. Just food for thought.

<https://agupubs.onlinelibrary.wiley.com/doi/full/10.1029/2023JC020651>

Response: *These are important points and ecologically interesting to consider in terms of potential importance of nitrogen fixation. To facilitate a deeper and more nuanced interpretation of the data, we have now provided additional calculations, data, and figures on biomass-independent specific nitrogen fixation rates, following Selden et al., 2021. See figure S9, Supplementary data sheet S1, lines 219-220, and 666-671.*

Minor comments

Minor comments:

[40]. "It is reported" language is awkward and unclear – Do you mean that you are reporting or someone else has reported?

Response: *This is now re-written.*

[47-50] Reporting the number of samples or stations in each region would help provide some context to the reported numbers. I recommend reporting a median for each region here for the same reason.

Response: *We have specified the number of stations in lines 46-47. We have not added the median but kept to reporting the range (from min to max) due to the word limit of the abstract.*

[78-79] This statement is vague – Are you specifically referring to the contribution of diazotrophs?

Response: *We have clarified this statement now, please see lines 76-77.*

[89] “pertinent need” language is odd. I recommend removing “pertinent”.

Response: *“Pertinent” has been removed, please see line 85.*

[173] Best practice for NH₄⁺ measurements is to measure within 2 weeks, even if filtered and frozen. The high potential for NH₄⁺ concentrations to have changed during storage should be acknowledged.

Response: *We agree and have now specified this uncertainty and associated care to be taken when interpreting the data in lines 155-157.*

[174] Were samples filtered before freezing?

Response: *Yes, this is now specified in line 154.*

[217] Is the value reported in atom%? How many replicates? Please report the full statistics, either here or in a table in supplementals.

Response: *Yes, it is in atom% as stated in line 198. More details have been added on this, please see lines 199-200.*

[223-225] Does this mean that the DCM samples from the first cruise were incubated under high light? If so, please add text noting this point and how it may have impacted results. If not, please add clarifying text.

Response: *Yes, this is correctly interpreted. Due to the more variable light conditions during the MIZ work (ranging from open water to pack ice), station-specific shading was applied. This is one reason for why the datasets are analysed partly separate through the manuscript. This is clarified now in lines 206-207 and 402-403.*

[229] pore size?

Response: *The details of this filter is specified upon its first mentioning in the text, please refer to line 186.*

[288-297] I applaud the authors for carefully considering and explaining these points. Nicely put.

Response: *Thank you!*

[421-426] Please report all relevant statistics here.

Response: Thank you for pointing this out. We have now included further statistical information, please see lines 393-400, and 417.

[488] Remove “is”.

Response: This has been removed.

[570] “This [the data] proposes” is awkward. I recommend either “Based on these results, we propose” or “The data suggest”.

Response: This has been re-formulated, line 494.

[585-589] Do your organic matter additions experiments provide any additional context here? If DOM release were driving the relationship between diazotrophic and ice-edge blooms, would you expect diazotrophs away from the ice-edge to respond to the addition of labile DOM?

Response: Thank you for the comment. This aspect is addressed in a later paragraph, please see lines 560-585.

[611-614] It’s unclear to me why this statement is given as if it were in conflict with the statements given above. Two ecologically disparate groups can stimulate activity of one another. I recommend rephrasing and removing the “acknowledge” language.

Response: This has now been reformulated, please see lines 534-536.

Reviewer 2

Summary

The authors report a study of diazotroph community composition and activity in the Arctic Ocean. Rates and studies of diazotrophy in the Arctic are scarce, so this report is valuable. The authors are well-qualified to carry out the work, the manuscript is well written, clearly organized, and the data are presented effectively. I believe the methods the authors used are robust. The authors speculatively make the case that as oceans warm and stratify it is essential to document the importance of N₂ fixation as a source of N to rapidly changing ecosystems such as the Arctic Ocean. Mostly the authors find low abundances and rates of N₂ fixation in the Arctic. However, the authors frame this result differently, e.g., in the abstract they claim N₂ fixation is "substantial" in the Arctic. I find this framing problematic given the low abundances and low rates of N₂ fixation that they measure. The field of marine N₂ fixation has suffered for decades because of reports of "significant" N₂ fixation when really, rates are low and are not put in a quantitative context. The field now has a problem, and I'm afraid this manuscript will further exacerbate the problem. While it's useful to document diazotrophy robustly, and I believe the authors have been careful with their methods, which I very much appreciate, it's still critical that the quantitative significance of N₂ fixation in an environment is placed in the appropriate, quantitative context. This includes adjusting statements that make it sound like 10% of NPP was supported across the Arctic by N₂ fixation, when it really was at a minor fraction of the stations where not even 10% of NPP was supported by N₂ fixation (see Table 2). It's also difficult to support the prediction that N₂ fixation rates will increase in the Arctic as it warms; here, the authors speculate.

Response: Thank you for your thorough and constructive review. We have revised the manuscript according to your comments. We are happy to hear that you find the results valuable and are thankful for your input on how to better frame the results. Specifically, we have:

- *Adjusted the phrasing of the importance of Arctic nitrogen fixation to better reflect the underlying data and specific environment (here: sea ice associated). E.g., we have revised the title, re-phrased the abstract (removed the word 'substantial', removed the statement of nitrogen fixation supporting up to 10% of local primary production), and throughout the manuscript been more careful with suggesting the importance of nitrogen fixation.*
- *Removed the prediction of increased nitrogen fixation from the abstract and kept it only in the discussion where we also problematize and give more context about the speculation, please see lines 673-683.*

Major comments

I would like to see the authors do an analysis similar to that in Turk-Kubo et al., 2013, Environmental Microbiology, doi:10.1111/1462-2920.12346, where they compare N₂ fixation rates estimated using ¹⁵N₂ uptake with diazotroph abundance estimated using similar nifH techniques described here, and literature-based estimates of per-cell N₂ fixation rates. Indeed, the authors reference this calculation, so I was surprised that they didn't do a similar calculation with their data. Showing that literature-based per-cell N₂ fixation rates for the diazotrophs they identify using molecular tools multiplied by the diazotroph abundances

they estimate are capable of generating the $^{15}\text{N}_2$ -based N_2 fixation rates they estimate would strengthen this study.

Response: Thank you for this suggestion. Due to the lack of cell-specific nitrogen fixation rates for all Arctic diazotrophs in question, we have not provided this calculation. Without relevant cell-specific rates, we do not believe such calculation is meaningful. The ecophysiology of reported Arctic NCDs are just starting to be explored, and can not be assumed to be similar to the few available per-cell nitrogen fixation rates. Furthermore, those estimated are associated with high variation which further problematizes the calculation. Finally, an intrinsic weakness in the calculation by Turk-Kubo is that the overall *nifH* gene abundance (and that of diazotroph cells) is unknown, because the qPCR only targets select groups. Furthermore, assuming that *nifH* copies equals diazotroph abundance is problematic (please see articles with doi numbers 10.1002/Ino.12036, 10.1002/Ino.12364, and 10.1002/Ino.12363).

Minor comments

-lines 175-177: total N and total P are not inorganic nutrient concentration analyses; please revise

Response: This has now been adjusted, please see line 157.

-lines 353-355: Please also report the NO_3+NO_2 and NH_4 concentrations at the same depth

Response: Please refer to Table 1 for all values. The parameters written out here are chosen due to their large contrast between the regions, as seen in Figure 2.

-line 367: Please clarify – if the N:P molar ratio you're reporting is water column inorganic nutrient concentration ratios, can you specify that? If it's something else (e.g., biomass or particulate stoichiometry? DON:DOP concentration ratios?) please indicate that; same on line 369

Response: This has now been specified, please see lines 351-352.

-line 383: I'm not sure what C:N ratio you're reporting here – if it's POC:PON, please specify. If it's other, please specify that, as well; same for Fig. 2 and Table 2 captions

Response: This has now been specified, please see line 368, fig 2 and table 2 captions.

-DON and DOP concentrations are reported in Table 1, but no methods are included for those analyses – please include methods for these measurements

Response: This has now been specified in lines 159-161.

-lines 421-424: Can you include the un-amended as well as DOC amended N2 fix rates for both Stn2 26 and 56 described here so the reader can get a sense for the magnitude of change in the rates?

Response: *This is now added, please see lines 394, 396-397, and 399-400. All data is also provided in Figure 3B and supplementary data sheet S1.*

-line 569: recommend changing “substantial” to “detectable”

Response: *The word substantial has been removed, please see line 492-494.*

Reviewer 3

Summary

This manuscript looks at N₂ fixation in the Arctic, a region where rates are low and variable. The manuscript is well-presented with clear figures and easy to follow text. I believe the lead author is a PhD student and they should be commended for assembling this well-formulated manuscript. My major and minor comments are listed below and they should be answered prior to publication. Well done on this nice piece of work.

Response: *Thank you very much for your nice words, and thorough and constructive review. We have revised the manuscript according to your comments. We believe your review has helped us improve the manuscript.*

Major comments

1) There is a big difference between the level of confidence displayed in the abstract compared to the level of confidence displayed in the conclusion about the results and their analysis. The discussion presented a more balanced consideration which was missing in the abstract. I encourage the authors to read Gu (2025) Evidence of limited N₂ fixation in the Southern Ocean. *Comm Earth & Env* 6: 264 which presents a balanced overview on the potential importance of N₂ fixation in polar environments.

Response: *Thank you for this comment. We have now more carefully adjusted the formulation, interpretation, and significance of the results in the title, abstract, and throughout the manuscript.*

2) It is well-known that rate measurements of ¹⁵N₂ assimilation have high variability, and can approach 20% variability. This makes the measurements challenging in the oceans where N₂ fixation is more prolific (5-10 nmol L⁻¹ d⁻¹). For marine environments where rates are close to detection limit, I think greater emphasis needs to be placed on improving and accuracy. This study uses deckboard incubations of three samples with no report of temperature levels during the incubation. The authors also rely on ¹³C-assimilation for estimating net primary production which is known to be less accurate than O₂-based methodologies (when conducted correctly). I would like to see Table 2 include the error propagated from both the ¹³C and ¹⁵N rate measurements, which will highlight the uncertainty associated with N₂ fixation in the Arctic.

Response: *Thank you for this important comment. We agree with the reviewer that it is important to provide all data and associated error propagation calculations, especially when reporting low rates from new environments. We have followed the latest recommendations put forward in White et al 2020 (doi: 10.1002/lom3.10353) and used their template for streamlined interpretation of raw data, limit of detection calculations, variability, and associated error propagation values (i.e. % or error for each parameter: time, atom percent of dissolved N₂, atom% of PON at T₂₄, atom% of PON at T₀, and the PON concentration), please see Supplementary data sheet S1. In this file, we have labelled each entry based on if they were above both LOD methods, only one, or below both.*

The limits of detection are additionally provided in the text (see lines 375-376, 381, and 387). More information is now given in lines 216-219, 384-385, and 387-391, and in the text to Table 2, we now refer to the Supplementary data sheet S1.

Regarding temperature in on-deck incubators, it was continuously recorded with HOBO loggers during cruise 1 (unfortunately not during cruise 2) and it has now been added in lines 203-204.

We are not aware of earlier studies providing a comprehensive error propagation analysis of ^{13}C -incorporation data. The availability of a recommended structure for such analysis for ^{15}N -incorporation data is thanks to a large common effort by the nitrogen fixation research community behind the White et al. review. Similarly, a standard error propagation scheme should be developed by specialists in measurements of primary production.

Please see our response further down regarding the choice of the ^{13}C method. We are not aware of literature documenting biases or particular uncertainty (more than other methods) advising against the ^{13}C incorporation method. If available, we would be happy to cite such studies to make readers aware that our ^{13}C data should be treated with some caution. Please see added information in lines 221-224.

Minor comments

- Abstract

Sentence 1 nitrogen already is an important determinant of primary productivity

Response: *We agree, but by using the word “increasingly” this is implied.*

Sentence 4 After you have used the chemical formula (N_2) in sentence 2 you need to use it thereafter

Response: *Thank you for noticing this. We have removed the chemical formula as we do not use it, please see line 43.*

Sentence 4 Data shortage impedes understanding nitrogen fixation supported productivity in the global oceans, not just Arctic

Response: *We agree. Due to shortening of the abstract, this part is no longer included.*

Is the positive correlation between N_2 fixation and primary production an auto-correlation? If yes, then should it be highlighted in the abstract as a key finding?

Response: *Our apologies, but we do not understand this comment.*

I struggle with the summaries of N_2 fixation and productivity/phytoplankton in the Central Arctic Ocean and Eurasian Arctic. For one site you mention rates of net primary production while at the other site you mention a phytoplankton bloom but no rate of net primary

production. Also if one site ranges from 0.4 to 2.5 nmol L⁻¹ d⁻¹ and the other site ranges from ND to 5 nmol N L⁻¹ d⁻¹, what is the difference?

Response: *We have now clarified this, please see lines 49-51.*

- Introduction

Line 61 – what do you mean by altered stratification? This is quite vague and it would be helpful to know if you mean weakened or strengthened

Response: *We chose the word altered as whether stratification increases or decreases is region-dependent.*

Line 83 I am very wary of over-exaggerating the importance of N₂ fixation in the Arctic.

Response: *We agree that its importance should not be exaggerated until solid data supports such claims. However, this sentence just states that nitrogen fixation is an overlooked process, as it was previously not thought to take place at all, which we now know that it does. We have re-formulated, please see line 80.*

- Method

Nice charts/locations

Response: *Thank you!*

Line 163 - Why did you target the DCM? What depths did other N₂ fixation studies target?

Response: *The reasoning behind the target depth is provided in lines 111-114. To see the depths of other nitrogen fixation studies, we kindly refer you to the supplementary material of von Friesen and Riemann, 2020 (doi: 10.3389/fmicb.2020.596426).*

Line 206 – Why would you use ¹³C for measuring net primary production? You cite Hama (1993) as the method, but that's 30 years old and methods have improved. The issue with ¹³C is its accuracy. There are much more reliable measurements using O₂-based methodologies.

Response: *We chose the ¹³C method to measure primary production as it is the current practice in the nitrogen fixation community, and the feasibility of that the incorporation of ¹³C is carried out in the same bottle incubations as ¹⁵N. Indeed, this is commonly performed alongside ¹⁵N-based nitrogen fixation rate measurements in oceanic systems, which facilitates comparison to other studies in the field (see original method by Slawyk et al., 1977, doi: 10.4319/lo.1977.22.5.0925) e.g.,*

- *western North Pacific (Wen et al., 2022, Science Advances, doi: 10.1126/sciadv.abl7564),*
- *Baltic Sea (Eigemann et al., 2019, Plos one, doi: 10.1371/journal.pone.0223294),*

- North Pacific subtropical gyre (Gradoville et al., 2021, *Limnology & Oceanography*, doi: 10.1002/lno.11423)

-

All our data is reported in Supplementary Datasheet S2.

Line 208 – Triplicate measurements are the minimum for replication and when there is high uncertainty maybe increasing this to 4 or 5 would be helpful?

Response: We agree with the reviewer that proper replication is important (minimum three) and it would be good with even more replicates. However, this was not feasible due to practical reasons (e.g. the size of incubator, processing time of samples, water budget available from Niskin bottles).

Why don't you deploy a sediment trap and look at the d15N signal of sinking material?

Response: This would be interesting but was not within the scope of this study. We performed the first measurements of nitrogen fixation in the Central Arctic and focused on the rates, exploring the environmental drivers, the diazotroph community, their expression, and quantification of key groups.

Line 220 – Did you put a temperature logger in the on-deck tank? Its very easy for on-deck incubators to end up being a few oC higher than the ambient water-column and I have seen a few case studies were it was 3-5oC higher!

Response: Regarding temperature in on-deck incubators, it was continuously recorded with HOBO loggers during cruise 1 (unfortunately not during cruise 2) and it has now been added in lines 203-204.

Line 235 – Thanks for providing data

Response: You're welcome, we agree this is important to the field.

I don't have comments about the sequencing methods

Response: Ok.

- Results

Figure 2 I don't understand this figure. Are you suggesting that the silicate and phosphate in multi-year ice is a driving factor? Why then did you get the phytoplankton bloom in the transect

Response: This figure is a visual representation of environmental differences between the study regions (ice regimes). It should be remembered that the CAO is covered by sea ice, whereas the MIZ is gradually going from open water to pack ice. Therefore, it is very likely phytoplankton are light limited in the CAO, thus allowing dSi to build up.

Does Figure 2 repeat information shown in Table 1?

Response: Thank you for the question. No. Figure 2 is a principal component analysis based on the data from Table 1, but showing the relationship between the different stations in terms of their environmental conditions. It is provided as a justification and explanation of how regional characterization was made (see the different shapes (circles, triangles and squares) in plot Figure 2).

Line 405 'Nitrogen fixation was detected under all sea ice regimes' doesn't this contradict what you stated in the abstract

Response: Please refer to Figure 3B. Nitrogen fixation was reported from all sea ice regimes, but just not along transect 1 before the phytoplankton bloom started.

Line 421 Does stimulation within 24 hours surprise you? How is the community able to respond so quickly? Does this reflect an increase in growth rates of bacteria or is nitrogenase activated in the wider microbial community

Response: The DOC mixture contained both labile and semi-labile monosaccharides which are expected to be rapidly utilized by the microbial community. The Arctic Ocean is characterized by refractory DOC largely originating from terrestrial sources, and thus a fresh source of DOC is expected to generate a response. We do not know what parts of the microbial community that are responding to the addition. The findings and interpretations we made on this matter are found in lines 560-585.

Figure 3A Why does DOC decrease primary production at Station 61? Please can this graph be labelled ¹³C-primary production

Response: We have added ¹³C in the figure legend. Station 61 is in the high-melt zone of transect 2, where the ice-edge phytoplankton bloom was ongoing, likely causing associated naturally high concentrations of labile DOC. Therefore, we do not think that our addition of DOC caused the difference seen. We note that the difference between the DOC treatment and the control is not significant ($p=0.05$ (not significant), $\chi^2=3.86$, $n=6$).

Figure 3B The $\leq 20\%$ variability in rate measurements is always tough (just an observation).

Response: Yes, we agree it is challenging with the high variability of the data.

- Discussion

The discussion is very long (>2400 words). You have 4 pages linking N₂ fixation to primary production alone.

Response: *The word limit is 5000 words (excluding methods), which we are about 500 words below. We have therefore not shortened the manuscript.*

What is the take-home message about NCDs. More variation in the MIZ – why is this?

Response: *The take-home messages about NCDs are:*

- *Our study supports the notion that NCDs are the main players in nitrogen fixation of the Eurasian Arctic Ocean*
- *NCDs are the likely key contributors to the measured nitrogen fixation rates*
- *Our data suggest that Beta-Arctic1 is a key NCD group in the Arctic Ocean*
- *We propose a ubiquitous distribution of the two NCD groups Gamma-Arctic1 (class unknown) and Gamma-Arctic2 (Oceanospirillales) at the DCM of the CAO.*

Please see lines 589-590, 603-605, 607, 625-626.

Line 607 Transect line or line of thinking

Response: *This has now been re-formulated, please see line 531.*

Line 608-614 Thank you!

Response: *You are welcome!*

- Supplementary Info
I appreciated the transparency of data

Response: *We are happy to hear that!*

To the editors

Dear Associate Editor Alice Drinkwater of Communications Earth & Environment,

Thank you for facilitating the second round of peer review of our manuscript “Nitrogen fixation under declining Arctic sea ice” (COMMSENV-25-1726-B). We are happy to see that all three reviewers support our study. We have carefully gone through and responded to each comment provided by the reviewers. We are pleased to now submit a revised version of the manuscript. Our responses are written in italics, and all changes can be traced through tracked changes in one of the manuscript files. Changes in response to reviewer comments are highlighted in blue. The other changes are editorial formatting requests.

We want to thank the reviewers for their qualified, thorough, and constructive comments.

Best regards,

Lisa W. von Friesen and Lasse Riemann, on behalf of all authors

Reviewer 1

I am satisfied that the authors have addressed my comments, and I appreciated their updated N₂ fixation calculations. I have only a few minor comments (associated with line numbers from the revised, marked up text):

[abstract] In line with comments from the other reviewers, I suggest adding some indication of error magnitude where Nfix rates are reported in the abstract. It helps contextualize the numbers for those who are interested in using them, but may not fully understand where they come from.

Response: *We are happy to hear you are satisfied with our revisions and would like to thank you for your valuable input. We now additionally provide the standard deviations in the abstract.*

[217] It would be useful for a casual reader here to specify why two LOD calculations were made and what the difference is.

Response: *A more detailed explanation of the two methods is provided in Supplementary Datasheet S1, along with the references cited.*

[222-224] I would point out to the authors that error propagation is standard practice across analytical chemistry and the template provided by White et al simply applies this practice in the context of N₂ fixation rates. In other words, there is nothing that need be “developed”. That said, I think it is okay not to do this for Cfix (as suggested by reviewer 3) because the authors already report standard deviation across replicates, which includes both analytical and biological variability in the measurement. Moreover, unlike with N₂ fixation, ¹³C-Cfixation rates generally have signals that are greater than analytical noise and thus easier to distinguish without analyzing sources of analytical error.

Response: *Thank you for this clarification. We have removed this sentence about ¹³C error propagation from the manuscript.*

Reviewer 2

I am mostly satisfied with the responses from the authors, thank you.

Lines 157-161: However, the authors still have not provided the methods they used to measure TDN and TDP concentration. Was TDN measured with DOC by high temperature combustion? Was persulfate oxidation or UV oxidation used? Similarly, for TDP, was the ash-hydrolysis method used? UV oxidation? Please specify the method used and provide references for the method, as well as precision of the method (i.e., +/- 0.6 μM TDN). Same for inorganic nutrient concentration measurements.

Response: *Thank you for your valuable review. We are happy to hear that you are satisfied with our responses. We now refer to the cruise report (line 512, ref 55) where the analytical methods are described in detail. For the reviewer, we have inserted the explanation below. In addition, we have also inserted the limits of detection for all nutrients: lines 505-506, 508-509, 512-514.*

During PS131, dissolved nutrients (PO_4^{3-} , $\text{NO}_3^- + \text{NO}_2^-$, $\text{Si}(\text{OH})_4$, total nitrogen and total phosphorus) were measured using standard colorimetric techniques with a segmented continuous flow nutrient analyser (AA500, Seal Analytical). The Seal Analytical manifolds to measure total nitrogen and total phosphorus are based on a combination of the wet oxidation method (e.g., Valderrama 1981, Solórzano and Sharp, 1981; Raimbault et al., 1999; Sharp, 2002) and UV oxidation method (e.g., Armstrong et al., 1966; Karl and Björkman, 2002), where the appropriate persulfate reagent is added to the sample pipeline right before entering a closed chamber through a quartz coil surrounding a 16 W UV lamp. In the case of TP, following the UV digester, the sample enters through a heater at 90°C. After the oxidation steps, the manifold continues as typical for $\text{NO}_3^- + \text{NO}_2^-$ and PO_4^{3-} . Precisions were equal or better than $\pm 0.10 \mu\text{M}$ for $\text{NO}_3^- + \text{NO}_2^-$, $\pm 0.004 \mu\text{M}$ for NO_2^- , $\pm 0.05 \mu\text{M}$ for $\text{Si}(\text{OH})_4$, $\pm 0.022 \mu\text{M}$ for PO_4^{3-} , $\pm 0.39 \mu\text{M}$ for TN and $\pm 0.021 \mu\text{M}$ for TP. The limits of detection were: 0.14, 0.006, 0.02, 0.01, 0.09 and 0.01 μM for $\text{NO}_3^- + \text{NO}_2^-$, NO_2^- , $\text{Si}(\text{OH})_4$, PO_4^{3-} , TN and TP, respectively. Further information about the Seal Analytical methods used during PS131 is reported in Kanzow (2023).

References:

Valderrama, J. C. The simultaneous analysis of total nitrogen and total phosphorus in natural waters. Mar. Chem. 10, 109–122 (1981).

Solórzano, L. & Sharp, J. H. Determination of total dissolved nitrogen in natural waters. Limnol Oceanogr 25, 751–754 (1980).

Raimbault, P., Pouvesle, W., Diaz, F., Garcia, N. & Sempere, R. Wet-oxidation and automated colorimetry for simultaneous determination of organic carbon, nitrogen and phosphorus dissolved in seawater. Marine Chemistry 66, 161–169 (1999).

Sharp, J. H. Analytical methods for total DOM pools. in (eds. Hansell, D. A. & Carlson, C. A.) 35–58 (2002). doi:10.1016/b978-012323841-2/50004-x.

ARMSTRONG, F. A. J., WILLIAMS, P. M. & STRICKLAND, J. D. H. Photo-oxidation of Organic Matter in Sea Water by Ultra-violet Radiation, Analytical and Other Applications. Nature 211, 481–483 (1966).

Karl, D. M. & Björkman, K. M. Dynamics of DOP. in (eds. Hansell, D. A. & Carlson, C. A.) 249–366 (2002). doi:10.1016/b978-012323841-2/50008-7.

Kanzow, T. (2023): The Expedition PS131 of the Research Vessel POLARSTERN to the Fram Strait in 2022 / H. Bornemann and S. Amir Sawadkuhi (editors) , Berichte zur Polar- und Meeresforschung = Reports on polar and marine research, Bremerhaven, Alfred-Wegener-Institut Helmholtz-Zentrum für Polar- und Meeresforschung, 770 , 320 p. . doi: 10.57738/BzPM_0770_2023

Reviewer 3

The authors have responded to the questions raised and I do not think the manuscript can be substantially improved. I recommend publication. Good work!

***Response:** We are happy to hear that! Thank you very much for your valuable input.*

1 **Manuscript ID:** COMMSENV-25-1726-T

2 **Brief article summary:** This study reports N₂ fixation rates and diazotroph abundances
3 in the under-sampled Arctic Ocean, with a focus on the relationship of polar diazotrophic
4 to sea ice. N₂ fixation rates were elevated in waters influenced by sea ice melt, which
5 the authors argue is likely due to nutritional coupling between heterotrophic diazotrophs
6 and primary producers.

7 **Comments on quality/impact:** N₂ fixation remains a notoriously difficult process to
8 model and, consequently, represents a critical area of uncertainty in Earth Systems
9 Models as marine primary production is largely nitrogen-limited and sets the upper limit
10 on the biological carbon pump. There is some burgeoning consensus within the field at
11 present that one of the primary factors causing this difficulty is the ecophysiological
12 diversity of diazotrophs. I believe that this study offers two important contributions to the
13 field: (1) It offers new data in a critically under-sampled and historically overlooked
14 region, and (2) it begins to draw a potential mechanistic connection between the activity
15 of presumably heterotrophic polar diazotrophs and productive waters in the marginal ice
16 zone.

17 The methods applied are robust and the study is appropriately positioned within the
18 context of pertinent literature. Some very minor editing could improve the clarity of the
19 text. (I've tried to help by pointing out a few examples of what I mean in my specific
20 comments below.)

21 **Major comments:**

- 22 • The impact of this article could be significantly increased by updating Sipler et
23 al.'s (2017) estimated contribution of the Arctic Ocean to global fixed N inputs
24 with the data reported herein as well as in other recent publications.
- 25 • The authors stress the potential importance of sea ice melt and ice margin
26 primary producers in stimulating diazotrophy. It's important to note, of course,
27 that mathematically N₂ fixation rates scale with PON, and PON generally scales
28 with primary production. It is thus not very surprising for somewhat higher N₂
29 fixation rates where PON is higher. We see this in coastal waters where higher
30 Nfix rates occur in high productivity waters even though Nfixers likely make up a
31 comparatively smaller portion of total community production (eg Selden et al
32 2024, link below). An additional point of interest here is that the relative
33 contribution of N₂ fixation rates (taken either as shown in Table 2 or by looking at
34 specific N₂ uptake rates as in article linked below) is elevated in CAO, basically
35 nil in MIZ Tr. 1, and low-moderate in MIZ Tr. 2. One could interpret this as
36 suggesting a greater niche for diazotrophs in the CAO relative to the MIZ, given
37 that they are contributing more substantially to N turnover under the local

38 conditions. Just food for thought.

39 <https://agupubs.onlinelibrary.wiley.com/doi/full/10.1029/2023JC020651>

40

41 **Minor comments:**

42 [40]. “It is reported” language is awkward and unclear – Do you mean that you are
43 reporting or someone else has reported?

44 [47-50] Reporting the number of samples or stations in each region would help provide
45 some context to the reported numbers. I recommend reporting a median for each region
46 here for the same reason.

47 [78-79] This statement is vague – Are you specifically referring to the contribution of
48 diazotrophs?

49 [89] “pertinent need” language is odd. I recommend removing “pertinent”.

50 [173] Best practice for NH₄⁺ measurements is to measure within 2 weeks, even if
51 filtered and frozen. The high potential for NH₄⁺ concentrations to have changed during
52 storage should be acknowledged.

53 [174] Were samples filtered before freezing?

54 [217] Is the value reported in atom%? How many replicates? Please report the full
55 statistics, either here or in a table in supplementals.

56 [223-225] Does this mean that the DCM samples from the first cruise were incubated
57 under high light? If so, please add text noting this point and how it may have impacted
58 results. If not, please add clarifying text.

59 [229] pore size?

60 [288-297] I applaud the authors for carefully considering and explaining these points.
61 Nicely put.

62 [421-426] Please report all relevant statistics here.

63 [488] Remove “is”.

64 [570] “This [the data] proposes” is awkward. I recommend either “Based on these
65 results, we propose” or “The data suggest”.

66 [585-589] Do your organic matter additions experiments provide any additional context
67 here? If DOM release were driving the relationship between diazotrophic and ice-edge
68 blooms, would you expect diazotrophs away from the ice-edge to respond to the
69 addition of labile DOM?

70 [611-614] It's unclear to me why this statement is given as if it were in conflict with the
71 statements given above. Two ecologically disparate groups can stimulate activity of one
72 another. I recommend rephrasing and removing the "acknowledge" language.